# Temporo-spatial cellular atlas of the regenerating alveolar niche in idiopathic pulmonary fibrosis

Praveen Weeratunga[1,2], Bethany Hunter[3], Martin Sergeant [4], Joshua Bull [5], Colin Clelland[6], Laura Denney[1], Chaitanya Vuppusetty[1], Rachel Burgoyne[3], Jeongmin Woo[1,4], Tian Hu[1,7], Lee Borthwick[3], James Shaw [8,9], Agne Antanaviciute[1,4], Andrew Filby[3], Helen Byrne [5,10], Andrew Fisher [8,9] & Ling-Pei Ho [1,2] ✉

Healthy alveolar repair relies on the ability of alveolar stem cells to differentiate into specialized epithelial cells for gas exchange. In chronic fibrotic lung diseases such as idiopathic pulmonary fibrosis (IPF), this regenerative process is abnormal but the underlying mechanisms remain unclear. Here, using human lung tissue that represents different stages of disease and a 33-plex single-cell imaging mass cytometry (IMC), we present a high-resolution, temporo-spatial cell atlas of the regenerating alveolar niche. With unbiased mathematical methods which quantify statistically enriched interactions, CD206[hi] macrophage subtype and an alveolar basal intermediate epithelial cell emerge as the most statistically robust spatial association in the epithelial and immune cell interactome, found across all stages of disease. Spatially resolved receptor–ligand analysis further offers an in silico mechanism by which these macrophages may influence epithelial regeneration. These findings provide a foundational step toward understanding immune–epithelial dynamics in aberrant alveolar regeneration in IPF.

Tissue regeneration and progenitor cell function rely on local microenvironments that provide optimal signals for adult stem cell proliferation and renewal[1,2]. Following injury, the healthy lung exhibits a remarkable capacity for repair, supported by a range of progenitor cells including type II alveolar epithelial cells (ATII), club cells, KRT5⁻ bronchoalveolar stem cells (BASCs), and the rarer airway-based KRT5[hi] basal stem cells[3]. In contrast, this regenerative potential is impaired and often dysregulated in chronic lung diseases such as idiopathic pulmonary fibrosis (IPF).

IPF is a chronic, progressive fibrotic lung disease with a poor prognosis. Approximately half of patients die within five years of diagnosis, and current treatments—limited to two antifibrotic agents—only slow disease progression[4]. The fibrotic response is thought to result from an exaggerated reaction to minor alveolar injuries, where activated fibroblasts expand the interstitium by depositing collagen. This altered architecture is accompanied by a significant immune cell presence[5] and abnormal epithelial regeneration resulting in bronchiolar-like cells rather than the thin type 1 alveolar epithelial cells[6,7].

[1]MRC Translational Immune Discovery Unit, Weatherall Institute of Molecular Medicine, University of Oxford, Oxford, UK. [2]Respiratory Medicine Unit, Nuffield Department of Medicine, University of Oxford, Oxford, UK. [3]Newcastle University Biosciences Institute, Newcastle upon Tyne, UK. [4]MRC Centre for Computational Biology, Weatherall Institute of Molecular Medicine, University of Oxford, Oxford, UK. [5]Wolfson Centre for Mathematical Biology, Mathematical Institute, University of Oxford, Oxford, UK. [6]Anatomic Pathology, Weill Cornell Medical College, Doha, Qatar. [7]Chinese Academy of Medical Science Oxford Institute, University of Oxford, Oxford, UK. [8]Newcastle University Translational and Clinical Research Institute, University of Newcastle, Newcastle, UK. [9]Institute of Transplantation, Newcastle upon Tyne Hospitals NHS Foundation Trust, Newcastle upon Tyne, UK. [10]Ludwig Institute for Cancer Research, Nuffield Department of Medicine, University of Oxford, Oxford, UK. ✉e-mail: Ling-pei.ho@imm.ox.ac.uk

Single-cell transcriptomic analyses have recently identified a population of abnormal alveolar epithelial cells with basaloid or bronchial-like features[8–10] which could hold the key to understanding why regeneration is aberrant in IPF lungs[10]. These alveolar basaloid intermediates (ABIs) are ectopically located in the alveolar lining, and possess transcriptomic profiles that distinguish them from normal alveolar and bronchial epithelial or stem cell populations[8,9]. Human alveolar organoids simulating IPF conditions suggest that ABIs differentiate into KRT5[hi]KRT17[+] basal cells, a trajectory that could underlie the formation of non-functional, honeycomb-like cysts characteristic of IPF lungs[11]. However, the factors driving this differentiation remain unknown.

We hypothesise that the diseased alveolar microenvironment—rich in inflammatory signals, growth factors, and cellular cross-talk—sustains ABIs and alters their differentiation toward a basal phenotype. Here, we examine the immune landscape, its temporal organisation, and spatial interactions within the regenerating alveolar niche as a prerequisite to delineating how immune cells might influence ABIs and regeneration[12].

Using lung biopsies from distinct regions representing progressive stages of the fibrotic process, we used a 33-plex imaging mass cytometry approach to assign both cellular identity and fixed spatial coordinates to every individual cell within the intact tissue architecture. This reveals an immune cell rich microenvironment in the regenerating alveoli, present across all stages of disease, dominated by monocytes, macrophages and CD4 T cells. We then applied unbiased mathematical methods to quantify the strength of cell–cell spatial associations and determined the cell types that are co-located at frequencies exceeding those expected by chance. Within the epithelial and immune cell interactome, the CD206[hi] macrophage subtype emerges as the only immune cells that are spatially co-located with alveolar epithelial cells across all disease stages. This finding, along with the temporo-spatial profiling of cellular composition and interactions during disease progression, and spatially resolved receptor–ligand analysis provide a foundational step toward understanding immune–epithelial dynamics in aberrant alveolar regeneration in IPF.

## Results

### Defining the temporo-spatial regenerating alveolar niche in IPF lungs

The overall study steps are shown in Fig. 1a–k and described in detail in Methods.

Lung biopsies were obtained from patients with IPF ($n = 7$ patients; 3 lung biopsies per patient) undergoing lung transplantation. Whole explanted IPF lungs were coronally sectioned and examined by an experienced thoracic pathologist to identify a lobe with macroscopic evidence of advancing disease stages. A 1x1 cm² biopsy was taken from each of three visually distinct lung areas representing sequential progression from early to advanced fibrotic disease [(Adv, or 'A'); less fibrotic lung (Early, or 'E') and the lung region in between (Intermediate or 'Intm/I') which represents the leading edge of disease] (Fig. 1b). These terms are relative to each other since the lungs are from patients with advanced disease undergoing transplantation. Disease-free tissue obtained from left upper lobe of un-used lungs from healthy organ donors ($n = 2$) was obtained for comparison (Supplementary Table 1).

Formal histopathological analysis (independent to macroscopic analysis of disease stage) showed an overall progressive increase in the amount of collagen, smooth muscle and a reduction in the numbers of fibroblastic foci as disease progressed from Early to Intm and Adv stages (Fig. 1l, and Supplementary Table 2). To address our aim of examining the immune interactome around regenerating alveolar epithelium, all ROIs (1 mm²) were selected by an expert thoracic lung pathologist and two pulmonologists to fit the following criteria—

histopathologically identified alveoli with (i) presence of cells and (ii) presence of type II alveolar metaplasia (oft-used as a feature of alveolar regeneration[13,14] and (iii) lack of bronchi and bronchioles (Fig. 1m). The 'regenerating alveolar niche' in this paper is defined as the area around type II alveolar metaplasia-containing alveoli. In total, 53 ROIs were selected ($n = 19$ 'E', n = 20 'I' and n = 14 'A').

Sections were stained with a 33-plex metal-tagged antibody panel (Fig. 1n, Supplementary Table 3), designed to capture the breadth of the immune cell profile, and to identify regenerating alveolar epithelium, including the 'ABIs' and basal cells described by Adams, Habermann and Kathiriya et al.[8,9,11]. Single cell segmentation, dimensionality reduction, cell clustering and annotation were performed according to the SpOOx 2.0 (Spatial Omics Oxford) pipeline, an extension of the SpOOx pipeline which we developed previously[15] (Fig. 1f–k and Methods). A total lung area of 61.89 mm² was ablated and 313,818 single cells were obtained.

With our step-wise and iterative annotation method (Supplementary Fig. 1), we identified 29 cell clusters—12 structural and 17 immune cell types (Fig. 1o and Supplementary Table 4). Detailed annotation decisions, including criteria, and likely cells for 'undefined'(UD) and 'adjacent' (ADJ) cell clusters are provided in Supplementary Fig. 1, and Supplementary Table 4.

This section characterises our definition of the alveolar regeneration niche and assigns both fixed location and cellular identity to every individual cell within intact lung architecture. This provides the foundation for interpretative analyses of cellular composition, spatial associations and interactions for this niche across different disease stage. It also demonstrates that alveolar regeneration niches are abundant and consistently present across all stages of disease.

### Cellular composition of regenerating alveolar niche in IPF

In total, five alveolar epithelial cell types were annotated in the regenerating alveolar niche (Fig. 2a–e). By morphology (type II alveolar metaplasia), presence of basaloid markers and using relative expression of ProSP-C, KRT5, EPCAM and KRT17/7 (in order of hierarchy of importance), three groups of cells were annotated as ABIs. These were classified as follows – 'ABI_a' (KRT5[neg-lo] ProSP-C[neg-lo]), 'ABI_b' (KRT5[lo-mid] ProSP-C[neg-lo]) and 'ABI_b-DC ADJ' (Fig. 2f, g). ABI_b-DC ADJ subcluster showed markers of ABI_b and were the only alveolar epithelial cells to show CD11c and low levels of CD45 expression in keeping with presence of adjacent dendritic cells (DC) (Fig. 1f, and Supplementary Fig. 2a). Here, ABI_b-DC ADJ cluster represents a spatial designation—reflecting proximity between epithelial cells and immune cells. An alveolar-based 'basal cells' (KRT5[hi] KRT17/7[hi] ProSP-C[neg]), and a cell cluster containing AT II cells ('ATII')(KRT5[neg-lo] ProSP-C[mid/hi]) were also identified (Fig. 2f, and Supplementary Table 4, Supplementary Fig. 2b).

We propose a link between our alveolar epithelial cells and points along the trajectory of ABI differentiation suggested by amalgamation of protein markers and transcriptomic data from Kathiriya, Adams and Habermann[8,9,11], shown in Fig. 2h. ABI_b has lower ProSP-C expression compared to ABI_a; so is likely to be later than ABI_a in the differentiation trajectory. This is also supported by slightly higher levels of KRT5 in ABI_b compared to ABI_a (Fig. 2f and h). Our manual matching analysis between our and published data (Supplementary Fig. 3) suggests that the alveolar basaloid intermediate states (i.e., our ABI_a, ABI_b; Adam's AB; Habermann's KRT5[-]KRT17[+] and transitional AT II; and, Kathiriya's ABI1 and ABI2) are a mixture of cells along an approximate trajectory of differentiation suggested by Kathiriya's organoid studies[11].

16 clusters of immune cells were identified, representing the breadth of adaptive and innate immune cells, including rarer, but relevant, specialised cells like Tregs and γδ T cells, and subclusters of monocytes and macrophages (Supplementary Table 4).

In terms of abundance, ABIs and basal cells formed a large proportion of alveolar epithelial cells in the regenerating alveolar niche

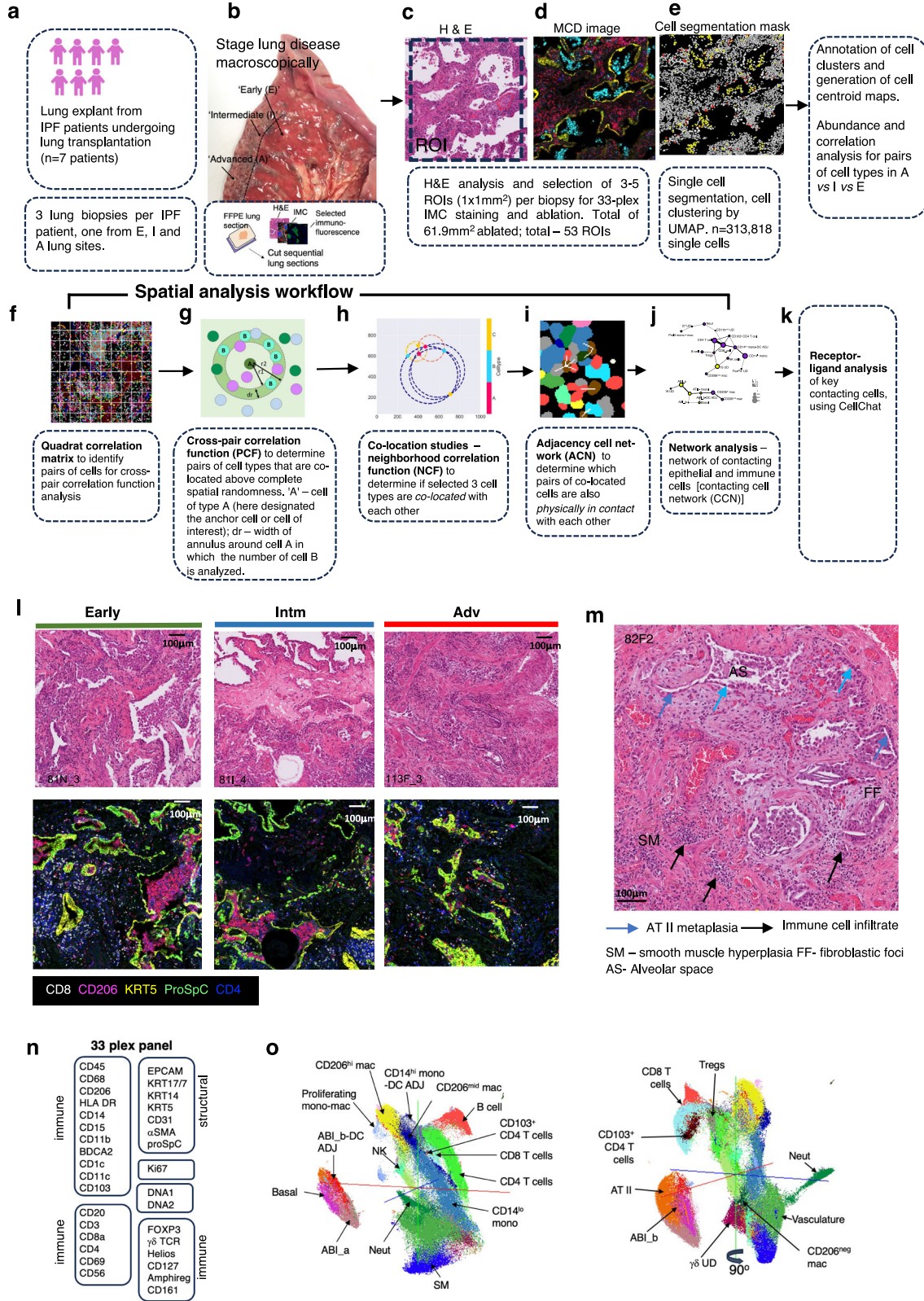

**Fig. 1 | Overall outline of study, the regenerating alveolar niche in IPF lungs and its cellular constituents. a–k** overall workflow of study from macroscopic selection of lung biopsy (**b**) to final analysis (**k**). **l** Representative ROIs from lung sections obtained in Early, Intermediate (Intm) and Advanced (Adv) lung regions (see **b**) showing histopathology features (e.g., collagen deposition, alveolar metaplasia, inflammatory infiltrate and smooth muscle hyperplasia) and matching MCD images stained with metal tagged antibodies—five of 33 markers shown. Lung sections are representative of $n = 19$ Early, $n = 20$ Intm and $n = 14$ Adv ROIs. Staining performed once. **m** Representative ROI from an H&E-stained IPF lung section showing criteria for ROI selection (presence of AT II metaplasia, immune cell infiltrate and lack of airways). **n** Metal-tagged antibody panel used for Imaging Mass Cytometry (IMC) staining (**o**). UMAP (front and back of 3 dimensional projection) of annotated cell clusters derived from 33 plex IMC staining. *H&E* hematoxylin and eosin, *FFPE* formalin fixed paraffin embedded, *AT II* type 2 alveolar epithelial cells, *Neut* neutrophils, *mac* macrophages, *mono* monocytes, *SM* smooth muscle, *ABI* aberrant basal intermediates. Hereon, where relevant, Early stage is depicted by green colour, Intm by blue and Adv by red.

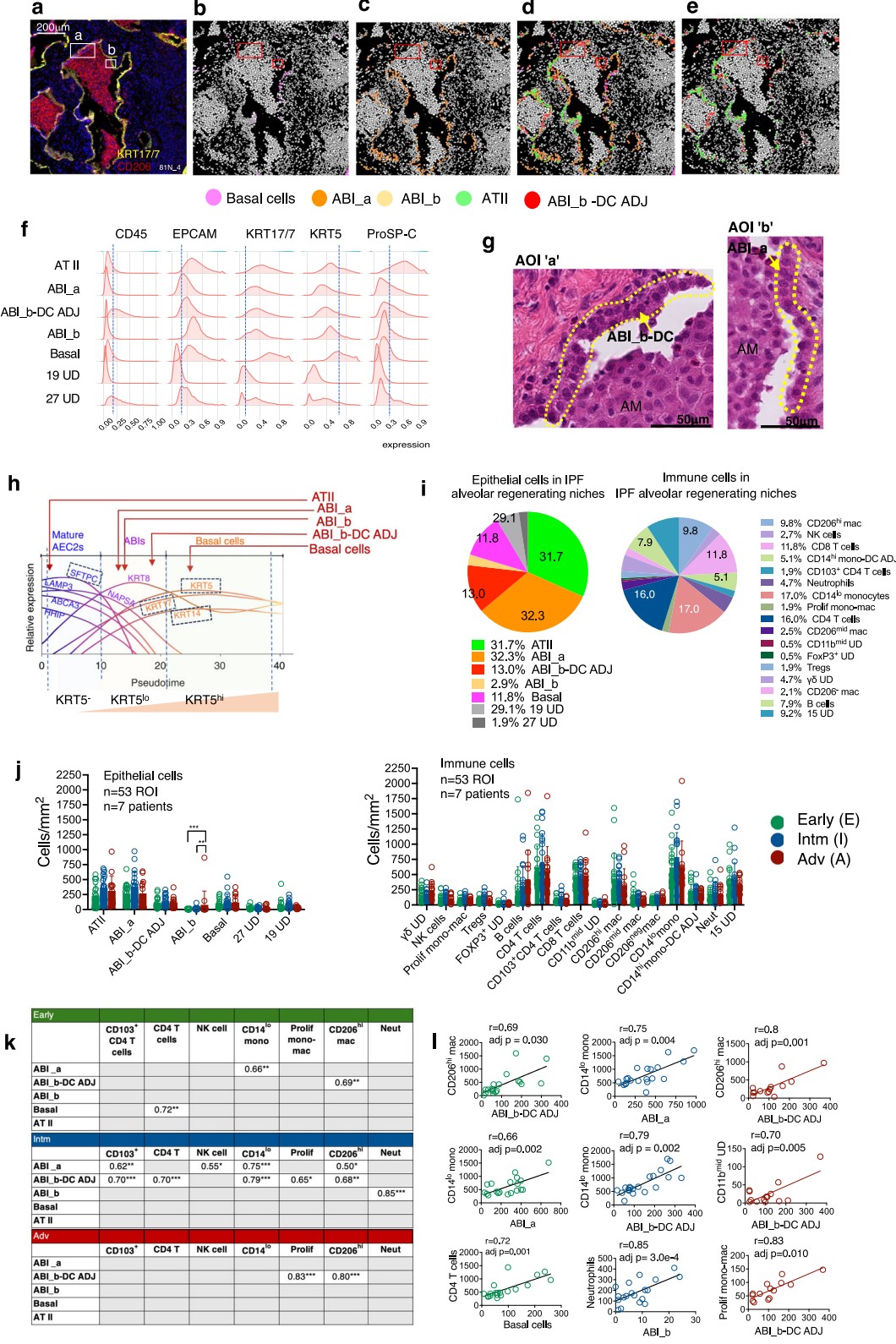

(up to 60%) (Fig. 2i left pie chart) (Supplementary Table 4). There was a near 1:1 ratio for ATII:ABI_a cell types.

Of the immune cells, CD14$^{lo}$ monocytes [17% of all immune cells] were the most abundant immune cells (Fig. 2i, right pie chart). These monocytes were distinct from CD14$^{hi}$ monocytes, both in location and association with other myeloid cells (Supplementary Fig. 4). CD14$^{lo}$

monocytes were scattered throughout the lung sections, co-located with endothelial cells, most likely from alveolar capillaries, making these most likely to be patrolling monocytes[16]. In contrast, CD14$^{hi}$ (classical) monocytes were found closely associated with DCs, forming the CD14$^{hi}$ mono-DC ADJ cell cluster, and were less abundant (5.6% of immune cells) (Fig. 2i and Supplementary Fig. 4b).

**Fig. 2 | Composition of alveolar epithelial and immune cells and their correlations in regenerating alveolar niche in IPF lungs. a** Representative ROI from Early disease stage showing KRT17/7 and CD206 protein expression (IMC staining). **b−e** Corresponding cell centroid maps for (**a**) showing location of ABIs, basal cells and AT II cells. No ABI_b is visible in keeping with their very low numbers. **f** Expression density plot for EPCAM, KRT17/7, KRT5, Ki67 and ProSP-C for all epithelial cell clusters in alveolar niche. Blue vertical lines represent background staining. ATII cells are the only cells with high ProSP-C expression; they also showed no/low KRT5 expression by gene expression analysis (see Supplementary Fig. 3b, c) and immunofluorescence staining (Supplementary Fig. 6). **g** High magnification of H&E images from 'a' and 'b' boxes in (A). These show that the morphology of ABI_a and ABI_b-DC ADJ cells are in keeping with formal histopathology feature of type II alveolar metaplasia. AM- alveolar macrophages. **h** Proposed points for ABIs, basal cells and ATII cells in the differentiation trajectory of abnormal alveolar epithelium as shown by Kathiriya's[11] human alveolar organoid studies. Broken blue boxes identifies markers used in our study. Diagram adapted from Kathiriya JJ et al.[11]; AEC 2 – type 2 alveolar epithelial cells. **i** Proportion (mean, for all 53 ROIs) of different epithelial and immune cells (as % of total epithelial and immune cells, respectively). **j** Number (cells/mm² of tissue) of immune (left graph) and epithelial (right graph) cells in Early, Intm and Adv disease stages. Bar plot provided for visualisation, showing mean and S.D. ***$p < 0.001$ is 7.2e-06 **$p < 0.01$ is 5.0e-03 [comparison derived using generalised linear model (GLM) model statistics, with area as a normalising factor]. **k** Correlation in abundance of cells (per mm² of tissue) between alveolar epithelial cell types and immune cells. Values are 'r' derived using Pearson's correlation (complete data in Supplementary Fig. 7). Only positive correlation shown; grey boxes−no significant correlation **$p < 0.01$; ***$p < 0.001$; Benjamini−Hochberg correction was applied for multiple comparisons. **l** Correlation plots for ABI-immune cell pairs with three highest r values for each disease stage (as shown in **k**). All analyses performed on data derived from 53 ROIs, $n = 7$ patients (see Fig. 1). Source data are available in Source Date File.

---

Macrophages can be divided into three subgroups according to CD206 and CD14 expression (Supplementary Table 4), reflecting their differentiation trajectory. CD206$^{hi}$ and CD206$^{mid}$ macrophages were found in the alveolar space whereas CD206$^{neg}$ macrophages were found in the interstitium (Supplementary Fig. 5). There was also morphological differences between CD206$^{mid-hi}$ macrophages and the tissue-based CD206$^{neg}$ macrophages (Supplementary Fig. 5b, c). CD206$^{mid}$ and CD206$^{hi}$ macrophages have a distinctive basophilic cytoplasm on H&E staining and were invariably found as tightly packed cell clusters in the alveolar space. These macrophage subsets also differed in their CD11c, CD1c and CD206 expression levels in keeping with different maturation stages (Supplementary Table 4).

Notably, neutrophils numbers (5.2%) were lower than monocyte and macrophage subsets (Fig. 2i).

In addition to myeloid cells, there was an unexpectedly high number of CD4 T cells (17.9 % of all immune cells), CD8 T cells (11.8%), and B cells (7.9%) (Fig. 2i). We detected CD103$^+$ resident CD4 T cells but not CD103$^+$ CD8 T cells [in keeping with prior findings that resident CD8 T cells are found predominantly in airways (rather than alveoli)][17].

There was no statistical difference in the overall proportion, distribution, or abundance of these immune and epithelial cells in ROIs from Early, Intm and Adv lung sections with the exception of ABI_b which was significantly more abundant in the advanced stages (Fig. 2j).

In contrast, lung sections from non-diseased donor lungs (HC) showed paucity of immune cells (as expected) (Supplementary Fig. 6a). Immunofluorescence staining of these healthy lung sections confirms scarcity of immune cells, and lack of ABIs in normal alveoli (Supplementary Fig. 6b−d).

A key finding from this part is that aberrant alveolar epithelium (ABI and basal cells) make up a large proportion (at least 64%) of the regenerating alveoli in IPF lungs. This means that very few cells are functioning ATI cells in regenerating alveolar niches in IPF (less than 5%; compared to 1:2 ATI:ATII in healthy lungs). Another biologically significant result is that immune cells in these regenerating niches were abundant even in the most advanced disease stage, dominated by CD14$^{lo}$ monocytes, CD4 T cells, CD8 T cells and CD206$^{hi}$ macrophage, and the composition of immune cells did not change significantly in the different stages of disease.

### Strong positive correlation in numbers of specific immune cells with ABIs in regenerating alveolar niche

Although there was no difference in the overall composition of immune cells between Early, Intm and Adv disease stages, we found striking and significant correlations in numbers between specific immune cells and alveolar epithelial cells at different disease stages (Fig. 2k, and Supplementary Fig. 7). The highest number of significant ABI-immune cell correlates were found in the Intm stage (Fig. 2k). Here, ABIs correlated positively with CD103$^+$ CD4 T cells, CD4 T cells, CD14$^{lo}$ monocytes, CD206$^{hi}$ macrophages, NK cells, proliferating monocyte-

macrophages and neutrophils (r between 0.50 to 0.85; $p < 0.001$ to 0.03)(Fig. 2k, l).

Notably, correlations in numbers between ABI and neutrophils and NK cells were found uniquely in Intm (disease leading edge) stage (Fig. 2k). CD206$^{hi}$ macrophages were the only cell type whose numbers increased with ABI numbers in all stages of disease (with ABI_b-DC ADJ) (Fig. 2k, Supplementary Fig. 7). There was only one statistically significant negative correlation between ABIs and alveolar epithelial cells −CD206$^{neg}$ macrophages with ABI_b-DC ADJ ($r = -0.71$; $p = 0.02$).

These findings suggest possible roles for specific immune cells (CD103$^+$ CD4 T cells, CD4 T cells, CD14$^{lo}$ monocytes, proliferating monocyte-macrophage cells, CD206$^{hi}$ macrophages and neutrophils) in the pathogenesis of ABIs or vice versa. It is noteworthy that all correlation in immune cell numbers with epithelial cells were with ABIs (none with AT II).

### CD103$^+$ CD4 T cells, CD206$^{mid}$ macrophages and CD206$^{hi}$ macrophages spatially co-locate with aberrant regenerating alveolar epithelial cells

We next determined, without a priori specification of neighbourhood or types of cells, which cell types were spatially associated in each ROI and compared this across the three disease stages. We used an updated suite of mathematical tools (SpOOx 2.0) to: (a) identify pairs of cells types that are more frequently co-located with each other than would be expected under complete spatial randomness (CSR) [via a 2-step sequential method - quadrat correlation matrix (QCM) then, cross pair correlation function (cross-PCF)] (b) determine whether three cell types are co-located with each other more frequently than under CSR, [using a mathematical function called neighborhood correlation function (NCF)] and (c) construct a network of contacting cells [via calculation of 'adjacency contact network(ACN) and then contacting cell network (CCN)] (Fig. 1f, j). This sequential pipeline is described in Methods.

The cross-PCF, $g(r)$, quantifies co-location between cells of type A and B by measuring how frequently cells of type B are found in an annulus of inner radius $r$ and width $dr = 10 \mu m$, centred around cells of type A, compared to expected in CSR (Fig. 1g)[18]. We compute the cross-PCF at a fixed inner radius of r = 20 μm (and width $dr = 10$ μm) as the diameter of any cell of interest in our study ranges from a mean of 8 μm (T lymphocytes) to 21 μm for macrophages (Fig. 1g). We also calculate the 95% confidence intervals around $g(r = 20)$, following the bootstrapping process described by Loh[19]. To provide greater detail of the spatial distribution of co-located cells, a '$g(r)$ plot' (Supplementary Fig. 8a) provides information for the entire range of radii from 5 to 300 mm and a 'topographical correlation map' ('TCM') allows visualisation of how spatial co-location between cells of types A and B changes across a single ROI (Supplementary Fig. 8b).

In total, 450 (from a possible 2187) pairs of cells passed the first step of analysis (QCM) and were submitted for cross-PCF analysis

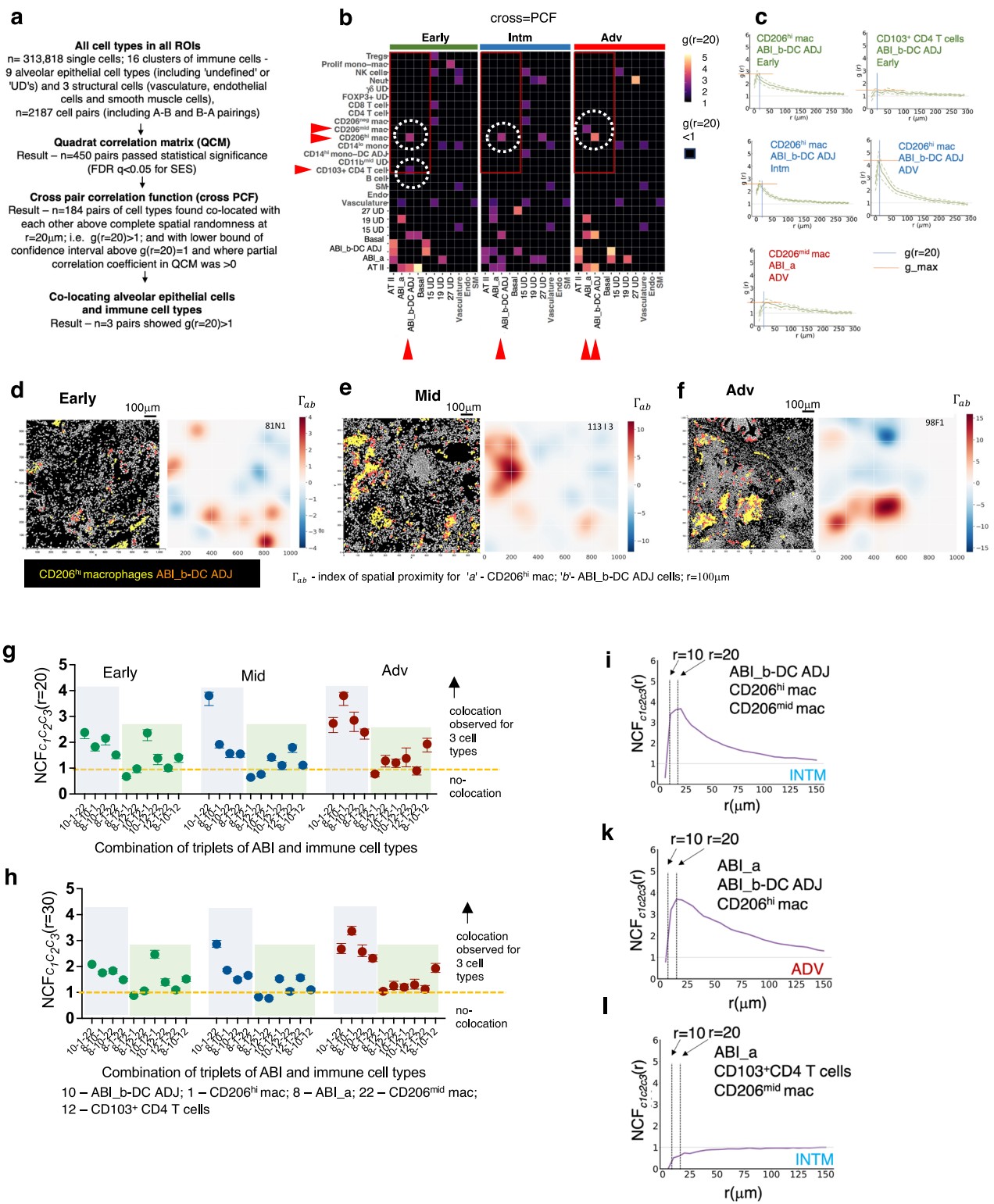

Γ_ab - index of spatial proximity for 'a' - CD206hi mac; 'b'- ABI_b-DC ADJ cells; r=100µm

10 – ABI_b-DC ADJ; 1 – CD206hi mac; 8 – ABI_a; 22 – CD206mid mac;
12 – CD103+ CD4 T cells

(Fig. 3a). Of these, 184 pairs showed co-location above expected at CSR [$g(r = 20) > 1$, (Source Data File)]. When these pairs were organised into disease stages, with epithelial cell types assigned as 'anchor cells' (around which the number of co-locating cells were counted), only three immune cell types were found to be co-located with any ABIs (in this case, ABI_b-DC ADJ and ABI_a) - CD206mid macrophages, CD206hi macrophages and CD103+ CD4 T cells (Fig. 3b, c). All other interactions were epithelial-to-epithelial cell type or immune-to-immune cell type co-locations. Notably, no immune cells were significantly co-located

with the ATII cell cluster, nor with the more differentiated ABI (ABI_b) or basal cells (Fig. 3b).

We next examined how co-localisation of immune cells varied with disease stage. CD206hi macrophages were found significantly co-located with ABI_b-DC ADJ at all disease stages. The only other significant co-locations were between early CD103+ CD4 T cells and ABI_b-DC ADJ cells in the Early stage of disease, and CD206mid macrophages with ABI_a in the Adv stage (Fig. 3b; distribution in strength of co-location shown in Supplementary Fig 9). Thus, despite highly

**Fig. 3 | Spatial co-location between aberrant regenerating alveolar epithelial cells and immune cells. a** Work flow and findings for each step of spatial analysis **b** Heat map showing g(r = 20) values for all structural vs all immune cell types. Red rectangular outline indicates g(r = 20) for alveolar epithelial vs immune cell types. Broken white circles and red triangles indicate alveolar epithelial - immune cell pairs with g(r = 20) > 1. Black squares - no co-location above complete spatial randomness [g(r = 20) < 1]. Full spatial data in Source Data File (tab for Figs. 3b, 4a–b, 5). **c** g(r) plots for the 5 co-locating alveolar epithelial-immune cell pairs from r = 0-300 μm. g_max = peak g(r) value in area around cell centroid of specified cell type from r = 0-300um; g(r = 20) = g(r) at 20 μm. **d–f** Cell centroid maps for CD206[hi] macrophage-ABI_b-DC ADJ pairs across the three disease stages, with their topographical correlation map (TCM). TCM visualises how the spatial correlation between two cell types changes across an ROI (here, ABI_b-DC ADJ and CD206[hi] macrophages). $\Gamma_{ab}$ scale indicates spatial proximity of cell type B to cell type A within a radius of r (in this case 100μm). High positive values indicate closer spatial

proximity compared to complete spatial randomness. **g, h** Output for Neighbourhood Correlation Function (NCF) which interrogates co-location of 3 (rather than 2) cell types. Here, triplets of ABI_b-DC ADJ, ABI_a, CD206[hi] macrophages, CD206[mid] macrophages and CD103[+] CD4 T cells (selected due to their co-location as pairs) are tested. Graphs shows $NCF_{C_1 C_2 C_3}(r)$ for all triplet ($C_1 C_2 C_3$) combinations of ABI (ABI_b-DC ADJ and ABI_a) to immune cells (CD206[hi] macrophages, CD103[+] CD4 T cells) at MEC radius of 20 μm (G) and 30 μm (H). Green pane – triplets containing CD103[+] CD4 T cells; blue pane – triplets without CD103[+] CD4 T cells. Error bars correspond to the 95% confidence interval around median, derived by bootstrapping for each displayed $NCF_{c_1, c_2, c_3}(r = 30)$, based on the number of MECs at the 5th and 95th centile. n = 53 ROIs from 7 patients. **i–k** Individual $NCF_{C_1 C_2 C_3}(r)$ plots for r = 0 to 150 μm for triplets that are most strongly spatially associated with each other compared to one which did not show association (**k**). Source data are available in Source Date File.

significant and strong correlation in abundance between ABIs and the immune cells, CD103[+] CD4 T cells and neutrophils at the leading edge of disease (Intm stage) (Fig. 2l, m), neither cell types were spatially associated with ABIs in this disease stage. g(r) plot for CD206[hi] macrophages - ABI_b-DC ADJ pairs showed the same qualitative distribution across the three disease stages (Fig. 3c)—the peak in co-location ('g_max') for CD206[hi] macrophages-ABI_b-DC ADJ cell pair was found near the centroid of ABI_b-DCs, indicating stronger co-location above CSR nearer to ABI. The highest values for g_max was observed in ROIs from the most advanced disease stage (Adv) (Fig. 3c), in keeping with the strongest co-location between the cell types. Examination of the individual cell centroid maps and topographical correlation map corroborated the g(r) plot findings (Fig. 3d–f).

Remarkably, despite their high numbers, CD4 and CD8 T cells, and B cells were not found significantly co-located around any alveolar epithelial cells (Fig. 3b). Similarly, although there were striking correlations in numbers between ABI cell types and CD14[lo] monocytes (Fig. 3b), there were no significant spatial associations between these cell types. The highest g(r = 20), as expected, were observed amongst alveolar epithelial cells (e.g., ABI_b-DC ADJ and basal cells) (Fig. 3b, Source Data Files).

We also detected co-locations between the vasculature cell cluster and some immune cell types (Fig. 3b) - NK cells and neutrophils were co-located around the vasculature cluster in Early disease stage, and CD14[lo] monocytes around vasculature in all disease stages. This is consistent with NK and neutrophil influx into areas of regenerating alveoli in the early stages of disease and CD14[lo] monocyte (purportedly patrolling monocytes) presence in the blood vessels at all stages of disease.

Altogether, these mathematical tools reveal statistically supported spatial changes in immune cell–ABI associations across the pseudotime inferred from the in-tissue linear progression of disease—relationships that would not have been captured by visual inspection or correlation analysis alone. Specifically, we found that CD206[hi] macrophages were co-located with ABI_b- DC ADJ cell clusters across all disease stages, while CD103[+] CD4 T cells were found co-located with this ABI earlier in disease. NK cells and neutrophils were found spatially associated with 'vasculature' in Early disease, and by the Intm stage, were found correlated in numbers with ABI_a and ABI_b respectively, though not co-located with them.

## CD206[mid] macrophages, CD206[hi] macrophages and ABI-DC cells form strongest spatially-connected foci at the leading edge of disease

Having identified CD206[hi] macrophages, CD206[mid] macrophages and CD103[+] CD4 T cells as the main immune cell types paired spatially with ABI_a and ABI_b-DC ADJs, we reasoned that these immune and ABI cell types could also be spatially connected with each other (beyond pairwise connections). To test this, we applied a mathematical method

which examines co-location of triplets rather than pairs of cell types[20]. The neighbourhood correlation function (NCF), $NCF_{C_1 C_2 C_3}(r)$, uses the concept of a minimum enclosing circle (MEC)—the smallest circle drawn which encompasses all three specified cells, of types $C_1$, $C_2$ and $C_3$ described in Methods and Supplementary Fig. 10). The smaller the radius of MEC, the closer the three cell types spatially (Supplementary Fig. 10). The NCF is a numerical representation of the strength of co-location of these triplets, obtained by comparing the distribution of MECs with radius $r$ against the distribution expected if all three cell types were distributed under CSR. Our relevant MECs are those with low radii but these radii have to be greater than the combined maximal diameter of the three cell types [which ranged from 8 μm (T lymphocytes) to 21 μm for macrophages] if they were found in closest proximity to each other (Supplementary Fig. 10). Accordingly, NCF for MECs with 20 μm and 30 μm were considered most relevant biologically. All combination of triplets of CD206[hi] and CD206[mid] macrophages, CD103[+] CD4 T cells, ABI_b-DC ADJ and ABI_a were examined.

We observed spatial co-location for 21 of 30 triplet combinations of ABI and immune cells, across all disease stages (Fig. 3g–j). The co-localisation was less pronounced (lower NCF) for triplets of CD103[+] CD4 T cells, macrophage and ABIs (Figs. 3g, h and k). All triplets containing macrophages and ABI subsets, and without CD103[+] CD4 T cells were co-located in all disease stages but most strongly in the Adv stage (NCF > 2, MEC with radii of 20 and 30 μm). The CD206[mid] macrophage, CD206[hi] macrophage and ABI_b-DC ADJ triplet at the leading edge of disease showed the highest NCF values (Fig. 3g–j).

These analyses complement and reinforce the spatial picture gained from our cross-PCF analysis. They confirm that among the cell pairs identified by cross-PCF as significantly co-locating, the individual cell types are also spatially co-localised with each other within defined regions. At the leading edge of disease (intermediate stage), the strongest spatial co-localisation was observed among ABI_b–DC ADJ cells, CD206[mid] macrophages, and CD206[hi] macrophages. In advanced disease, prominent co-localisation involved ABI_b–DC ADJ, ABI_a cells, and CD206[hi] macrophages. These findings suggest that, particularly at the intermediate stage, cellular interactions in the regenerating niche may centre on CD206[mid–hi] macrophages, dendritic cells, and ABI_b cells—highlighting a potentially critical immune–epithelial axis at this transitional disease stage.

## Neighborhood analysis for ABI cells supports involvement of CD206[hi] macrophages in differentiation of ABI cells to basal cells

To provide a complete picture of the neighborhood of ABIs, we collated all cell types (i.e., not just immune cells) that are significantly co-located with ABI_a and ABI_b-DC ADJ, revealing two further information about these ABIs. Firstly, ABI_a cells (purportedly, the earlier cells in the differentiation trajectory) are co-localised with the 'vasculature' cell cluster in Early and Intm stages [albeit a relatively weak co-location; g(r = 20) = 1.1]. Basal cells are found with ABI_a in the Adv stage

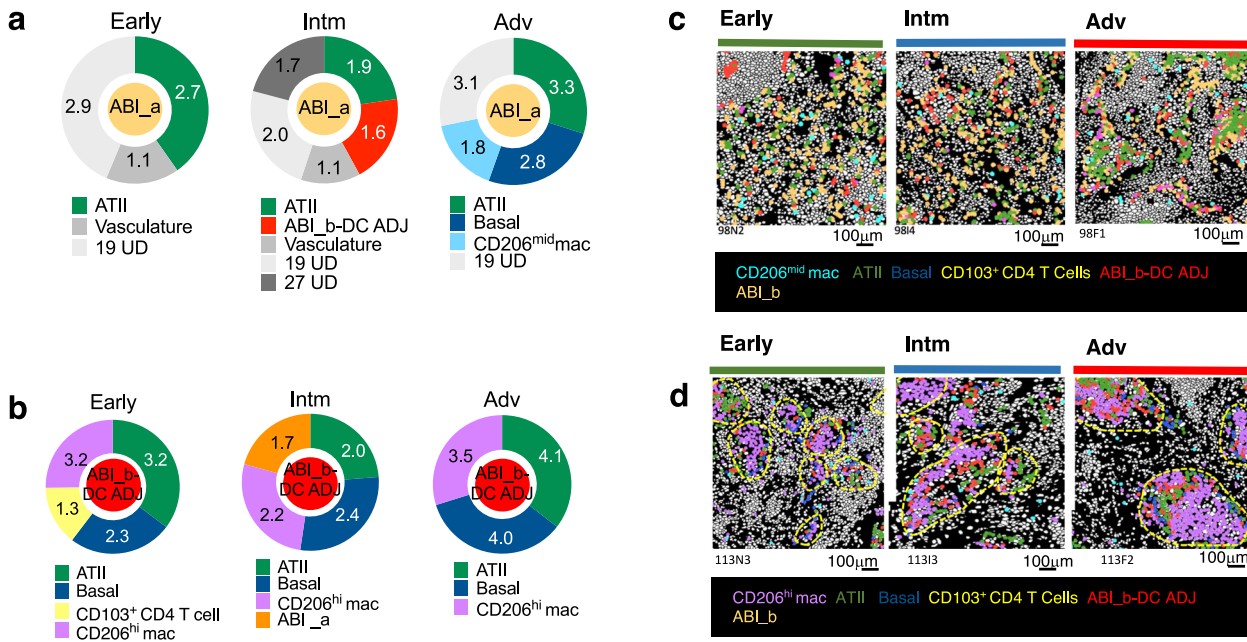

**Fig. 4 | Cellular neighborhood of aberrant regenerating alveolar epithelial cells. a, b** Donut charts show g(r = 20) values for all structural, epithelial and immune cell types that are co-located above CSR with ABI_a cell type (**a**) and ABI_b-DC ADJ cell type (**b**) for the different stages of disease. Size of segment is proportional to the relative value of g(r = 20). **c, d** Cell centroid maps showing the location of all cell types depicted in A and B in one representative ROI from Early, Intm and Adv for ABI_b (**c**) and ABI_b-DC ADJ (**d**). Colour of the different cell types are matched to the colour of the cell types in the donut charts in (**a**) and (**b**). Source data are available in Source Date File.

only, together with CD206^mid macrophages (Fig. 4a and c). In contrast, ABI_b-DC ADJ cell types form foci of cells with CD206^hi macrophages and basal cells from Early to Adv stage (Fig. 4b and d).

The differencein neighbourhoods of these two ABIs supports the concept that they represent distinct foci of aberrant alveolar regeneration. Co-localisation of CD206^hi macrophages and basal cells around ABI_b−DC ADJ clusters but not ABI_a cells, suggests that ABI_b−DC ADJ clusters are more mature cellular entities and supports a potential role for CD206^hi macrophages in promoting or sustaining ABI differentiation toward a basal cell fate.

**Network analysis of contacting cell pairs shows significantly more CD206^hi macrophages in direct contact with ABIs and reveals networks of immune and aberrant epithelial cells which evolves as disease progresses**

In the final part of the spatial analysis, we questioned which pairs of cell types are in direct physical contact [rather than merely in higher abundance within a radius of 20 μm around these cells, [as shown by the cross-PCF, g(r = 20)], since this will enable us to infer a potential receptor-ligand interaction.

To do this, we used the cell segmentation masks to identify co-located cells that are also in direct physical contact with each other, as previously described (we termed this the Adjacency cell network or ACN)[15] (Supplementary Fig. 11). In an extension to the ACN, for each cell type pair (e.g., cell types A and B), we then calculate the proportion of type A cells that is in contact with at least one of cell type B, and then the z-score [i.e., the deviation of the number of A- B contacts observed in our tissue compared to CSR (CSR here is derived by shuffling all cell type labels computationally by 1000 permutations, without moving the cell positions in the cell mask)]. A p value is computed for the z-score associated with each pair of cell types. The data are then filtered to retain pairs of cell types with g(r = 20) > 1, positive z-scores and p < 0.05 to create a network in which each connection ('edge') between cell pairs encompasses these three calculations. To differentiate this

final network from the ACN (which only determines which cells are in contact with each other), we termed this network the contacting cell network (CCN). See also Supplementary Information, 'Extended description on role of different spatial analytical method'.

CCN revealed that two immune cell types are in direct contact with ABIs above CSR - CD206^hi macrophages and CD206^mid macrophages, with ABI_b-DC ADJ and ABI_a cells respectively(Fig. 5a−c). CD206^hi macrophages-ABI_b DC ADJ pairs were found in direct contact across all disease stages while CD206^mid macrophage- ABI_a pairs were only observed in the Adv disease stage, adding and corroborating findings from the neighbourhood analysis (Fig. 4a−b).

Additional cell networks, associated with distinct disease stages, were also uncovered by the CCN. In the Early disease stage, ABIs, basal cells and ATII were found in direct contact with each other (Fig. 5d). CD206^mid and CD206^hi macrophages were the only immune cells found within this alveolar epithelial cell community. All other immune cells in the Early disease stage were spatially co-located forming a separate 'community' of cells (Fig. 5d). The pattern of spatial co-location changed at the leading edge of disease (Intm), where CD206^hi macrophages appear to be the central spatial connector for the ABI and immune cell communities (Fig. 5e), linking the ABI network to the immune cells. The highest numbers of directly contacting pairs of cell types were also found at the leading edge (Intm stage). This mega ABI-immune networks was not seen in Adv (Fig. 5f). Here, the immune cell community split into two main sub-CCNs: a myeloid-dominant network (comprising CD206^hi macrophages, CD14^hi monocyte-DC ADJ cluster, proliferating monocyte-macrophage cluster and CD206^mid macrophages), a T cell-dominant network (comprising CD8 T cells, CD4 T cells, CD103^+ CD4 T cells, Tregs, CD11b^mid UD and CD14^lo monocytes) (Fig. 5f). Persistence of a myeloid-dominant network with the ABI network in the Adv disease stage, signifies the co-existence of an active network of immune cells with aberrant alveolar epithelial cells, even at the most advanced stage of fibrosis. Of note, while co-location of CD103^+ CD4 T cells with ABIs was identified by cross PCF

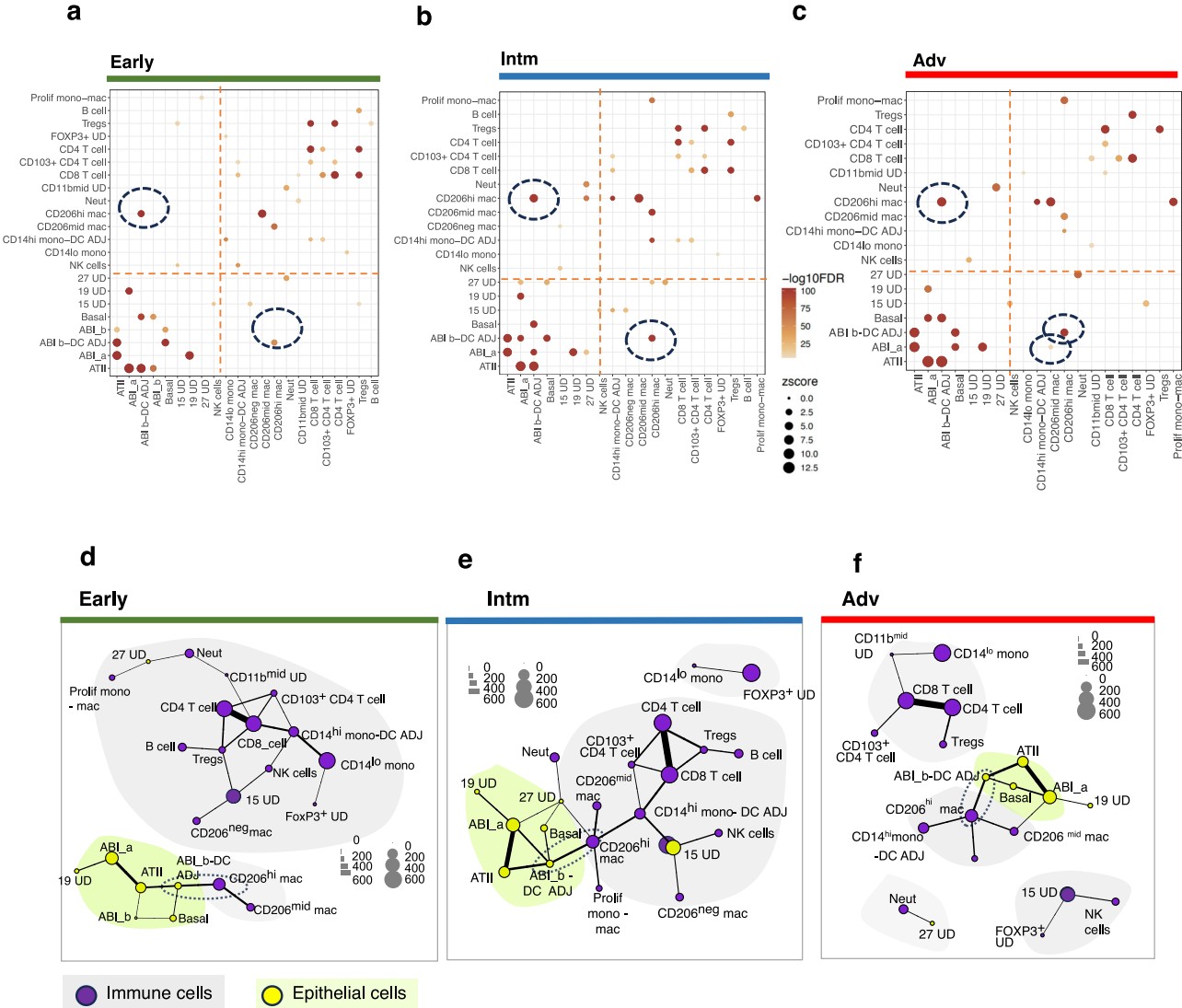

**Fig. 5 | Cell contact networks. a–c** Output for Cell Contact Network (CCN) for alveolar epithelial -immune cell pairs, depicting pairs of cell type A (here. alveolar epithelial cell types) that are in contact with at least one of cell type B (here, immune cell types) at a frequency greater than expected under complete spatial randomness (CSR), and where z score between observed and random contact has $p < 0.05$ (shown as bubbles). Broken circles Identify pairs of alveolar epithelial cell-immune cell types (we have not included pairs containing _UD cell types). Immune-immune or epithelial- epithelial pairings. Pairs with 'UD' cell clusters are not circled. Left upper quadrant - 'anchor cells' are alveolar epithelial cell; right lower quadrant output mirrors the left upper quadrant but 'anchor cells' are immune cell types. **d–f** CCN network for all cell type pairs showing 'communities' of directly contacting cell pairs (grey and lime coloured spaces). Connecting lines join cell type pairs that are in direct contact with each other with greater frequency than complete spatial randomness, and with z scores that have FDR q values < 0.05. Size of nodes represents number of cells per mm², thickness of lines represents number of contacts between cell types A and B. Grey spaces−communities of immune cells (purple nodes); lime spaces−communities of alveolar epithelial cells (yellow nodes). Note we have included the UD's in the network to show all cell types. Broken circles highlight ABI_b-DC ADJ and CD206hi macrophage pairs, which at all stages, link up aberrant alveolar epithelial communities with immune cell communities. Source data are available in Source Date File.

and NCF analyses, they were not found to be in direct contact with the ABIs, suggesting that any cross talk is likely to be via humoral communications.

In summary, analysis of contacting cell types at pair-wise level shows significantly more CD206hi macrophages in direct contact with ABIs than CSR, and they (and to a lesser degree, the CD206mid-ABI_a pairings in Adv stage only) were the only pairs of cell types that achieved this. In addition, the network analysis of contacting cell pairs reveals networks of immune and aberrant epithelial cells which evolved temporally as disease progresses. Importantly, at the leading edge of disease, CD206hi macrophages connect communities of immune cells with communities of aberrant alveolar epithelial cells.

These findings elevate the potential importance of CD206hi macrophages and ABIs interaction and prioritises ligand-receptor cross talk between these two cell types.

**Fibronectin and MIF signalling predicted as key signalling pathways between CD206hi macrophages and ABIs**

In the last part of our study, we explored the potential functional significance of direct contact between CD206hi macrophages and ABIs. To do this, we obtained publicly available single cell transcriptomic data (which provides depth in cell identity but not information on location or spatial interaction) from Habermann et al. [single cell RNA sequencing data on explanted lungs, from six groups

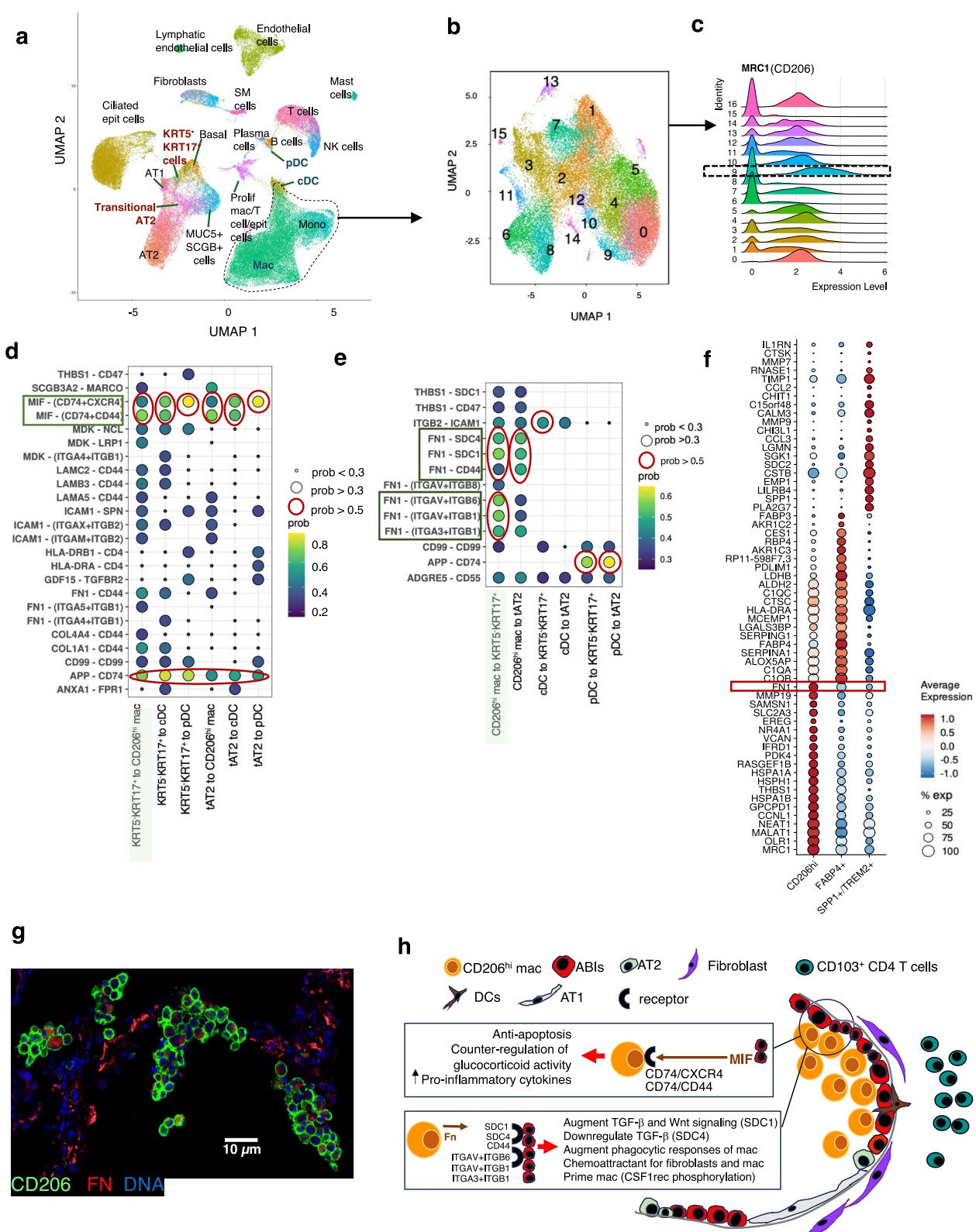

of patients, with lung diseases including IPF (*n* = 12), and non-fibrotic controls from lung donors (HC; *n* = 12)][9]. We chose Habermann's data as some of our trajectory analysis of ABIs was based on Kathiriya's organoid studies[11], and they matched their findings with Habermann's transcriptomic dataset[9].

We extracted the single cell metadata from the IPF and HC datasets, and first, re-clustered monocyte and macrophage clusters

(Fig. 6a). This was then resolved transcriptomically to identify a subset with a clear, discriminating level of CD206 expression (which we termed CD206hi macrophages) (Fig. 6b, c). Habermann's KRT5⁻KRT17lo ABI subset and transitional AT II subset were extracted for analysis, the former having been matched to our ABIs (Supplementary Fig. 3), and the latter as a comparator. Habermann's dataset did not allow determination of whether their 'basal cells' were the airway-based basal stem

**Fig. 6 | Receptor-ligand analysis between aberrant basaloid intermediates and CD206[hi] macrophages. a** Data extracted from of single cell RNA sequencing of lung digest cells from IPF (*n* = 12) and healthy control (*n* = 12) lungs deposited by Habermann et al.[9] reproduced here to demonstrate starting point for further clustering of macrophages. This UMAP retains Habermann's annotations. Text in red identifies Habermann's aberrant basaloid intermediate cells (KRT5[-]KRT17[+] cells) and transitional AT II cells, and in blue the two clusters of DCs. Monocytes (Mono) and Macrophage (Mac) cell clusters in broken blue circle are re-clustered (using 0.7 resolution in Seurat) to discriminate levels of CD206 (MRC1) gene expression and where there is a clear CD206[hi] subcluster (subcluster 9) shown in (**b**) and (**c**). Cluster 9 (in broken box) (**c**) showing the highest CD206 expression is annotated as CD206[hi] macrophages. **d–e** Receptor-ligand analysis using CellChat, with ABI as sender and CD206[hi] macrophages, cDC and pDC as receivers (**d**) and vice versa (**e**).

Red circles - highest probability of communication (>0.5); large bubbles indicate probability of communication > 0.3, and small bubble < 0.3; no bubble–no communication detected. Colour of the bubble represents levels of communication probability. *Y* axis – ligand-receptor; *x* axis ABI - immune cell pairs. tAT2–Transitional ATII; 'prob'–communication probability (**f**) Marker gene list for Cluster 9 (CD206[hi] macrophages) compared to the established macrophage subclusters (FABP4+ macrophage and SPP1+ macrophage)[8,24,25] found in IPF. Note that cluster 9 (CD206[hi] macrophages) shows a distinct difference in fibronectin (FN) expression compared to FABP4+ macrophage and SPP1+ macrophage subsets. **g** Confocal picture of immunofluorescent staining of FN in CD206[+] macrophages in one of our IPF lungs (one of 16 ROIs shown, *n* = 3 patients). **h** Schema proposing location of ABIs and CD206[hi] macrophages, and functional consequences of receptor-ligand engagement. '*MIF*' - macrophage migration inhibitory factor, '*mac*' - macrophages.

cells or the alveolar-based basaloid cells which were ectopically sited in alveoli. Therefore, we did not include this cell type in the analysis. As we had a population of ABI closely associated with DCs (ABI_b-DC ADJ), we also extracted the plasmacytoid DC (pDC) and classical DC (cDC) populations from the IPF dataset. The CD206[hi] macrophages, pDCs and cDCs were then submitted for receptor-ligand analysis using CellChat[21]. We directed CellChat to examine interactions between these three immune cell types and KRT5[-]KRT17[lo] ABI and transitional AT II in both directions (e.g., with CD206[hi] macrophages as senders (ligand) and ABI as receivers (receptors) and vice versa).

Communication probabilities (calculated in CellChat based primarily on receptor-ligand gene expression profile)[21] identified several receptor-ligand pathways between ABIs and their directly contacting immune cells. Amongst these, those with 'high' communication probabilities (defined by us, as those higher than 2 S.D. from the mean communication probability value) were (a) macrophage migration inhibitory factor (MIF) (as the ligand produced by ABIs) that engages the CD74-CD44 and CD74-CXCR4 complexes on CD206[hi] macrophages, and CXCR4 on DCs (Fig. 6d, b) fibronectin (as a ligand produced by CD206[hi] macrophages) that engages with integrin (ITGA3, ITGAV, ITGB1 and 6), Syndecan (SDC1 and 4) or CD44 receptors on ABI (Fig. 6e). Identification of APP (amyloid beta precursor protein), a cell surface receptor is intriguing, with very little known apart from its role in Alzheimer disease and amyloidosis[22,23].

To place this CD206[hi] macrophages cluster in the context of known macrophage subclusters in IPF, we re-interrogated the transcriptome of the macrophages in Habermann's dataset and identified the two established macrophage subsets found in single cell analysis of IPF lungs - the FABP4[+] macrophages [purportedly resident alveolar macrophages[24]] and the pro-fibrogenic SPP1[+]TREM2[+] macrophages[8,24,25] (clusters 0 and 1 respectively in Fig. 6c). CD206[hi] macrophages (cluster 9) were distinct from these two macrophage subsets and, interestingly, expressed conspicuously higher levels of fibronectin (Fig. 6f). We validated this FN expression by immunofluorescent staining in our IPF samples and found differential presence of FN expression on CD206[+] macrophages, supporting the transcriptomic findings (Fig. 6g). The potential functional consequence of receptor-ligand engagement is summarised in Fig. 6h.

Our data suggest that the CD206[hi] macrophage subset identified in our IPF lung tissue is the CD206[hi]FN[hi] alveolar macrophage subset in Habermann's lung digest dataset. This CD206[hi]FN[hi] alveolar macrophage is distinct from previously established resident and pro-fibrotic macrophage subsets in IPF lungs. Our in silico analysis predicts fibronectin and MIF as potential ligands in the interaction between CD206[hi] macrophages and ABIs.

## Discussion

In this study, we present the largest single-cell spatial dataset to date of regenerating alveoli in fibrotic human lungs, offering new insight into immune–epithelial interactions during alveolar regeneration in IPF.

Using lung tissues from well characterised stages of disease and precisely defined regions of interest (ROIs), we captured a spatially anchored snapshot of alveolar repair across the continuum of disease progression. Importantly, our analytical pipeline was unbiased by prior assumptions, and systematically interrogated the entire cellular landscape to identify statistically enriched spatial associations between all cell types.

The different spatial analytical methods add strength to each other and provide different information. Thus, the pair-wise comparison provided by the cross-PCF (Fig. 3b) identifies the pairs of cell types that are spatially associated with each other more frequently than would occur by chance. Triplet comparison provided by the NCF (Fig. 3g–k) identifies triplets of cell types that are physically associated with each other and the neighborhood analysis in Fig. 4 brings together all pairs of cell types that are spatially co-located with our cell type of interest (see also Supplementary Information).

We show that just over half (51%) of the alveolar epithelium in regenerating regions is composed of aberrant basaloid intermediates (ABIs). The surrounding immune landscape is numerically dominated by CD14[lo] monocytes, CD206[hi] alveolar macrophages, B cells, and CD4[+] and CD8[+] T cells. While the overall proportions of epithelial and immune cell types remain relatively constant across disease stages, their spatial organisation shifts. In early or less fibrotic regions, resident CD103[+] CD4 T cells are prominently associated with ABIs– appearing in pairwise interactions, in triplet associations with CD206hi macrophages and ABI_b-DC ADJ clusters, and as co-located neighbours within the immediate microenvironment of ABI_b–DC ADJ cells. As disease progresses, these interactions are replaced by spatial associations between CD206[mid] and CD206[hi] macrophages and both ABI_a and ABI_b–DC ADJ cell types. Notably, these regions frequently contain co-located basal cells, consistent with a trajectory in which ABIs differentiate into basal cells during the transition from early to advanced disease.

A key outcome from our analysis is the co-location of specific immune cells with aberrant basaloid intermediates but not any other epithelial cells. This means that ABIs are likely to be key points in the differentiation trajectory where immune cell cross talk occurs, and a priority target for therapeutics. ABI cells are more prevalent than hitherto known[8,9], and found in all regenerating alveolar epithelium, including those in advanced disease where there is scarce fibroblastic foci and maximal collagen deposition in the interstitium. This means that even at the point of severe end stage disease, alveolar epithelium is attempting to regenerate, and ABIs are still interacting with CD206[hi] macrophages; possibly providing opportunities for therapeutically targeting even at this late stage to prevent progression to basal cells.

Our receptor–ligand analysis advances existing approaches by incorporating spatial context, restricting interrogation to cell types that are physically co-located within the tissue. This spatial constraint significantly enhances the biological relevance and precision of predicted interactions, in contrast to conventional single-cell

transcriptomic methods that infer receptor–ligand pairs from all theoretically possible combinations without spatial resolution. Of the several receptor-ligand signalling pathways identified by CellChat, a highly relevant one is the FN- $\alpha_v\beta_6$ integrin pathway. Integrins are a family of cell adhesion molecules which mediate many key cell–cell and cell-matrix interactions during fibrosis[26]. They are a major node of communication between the stroma, immune cells and fibroblasts and closely involved in the initiation, maintenance and resolution of tissue fibrosis. $\alpha_v\beta_6$ integrin is required for fibrosis, and has been a therapeutic target in several clinical trials in IPF[27,28]. It is specifically found on epithelial cells and upregulated after injury, activating TGF-β, a key profibrogenic cytokine[29] which activates fibroblasts and facilitates epithelial-mesenchymal transition (EMT). Very little is known of FN or $\alpha_v\beta_6$ integrin's impact on the differentiation trajectory of ABIs but here, a possible consequence is the alteration of the epithelial regeneration and downstream effect on fibroblastic activity. Another receptor-ligand signalling pathway that may be relevant to ABI regeneration is the syndecan-1(SDC1)-FN-mediated ABI-macrophage cross talk. SDC1 is involved in regulating epithelial cell migration and adhesion[30,31] and could augment Wnt signalling[32]. In acute lung injury soluble Syndecan -1 mediates chemokine mobilisation, and influx and activation of neutrophils, whose numbers are found to correlate with ABI_b-DC numbers at the leading edge of disease (Fig. 2k–l)[31,33,34]. The converse cross-talk, with production of macrophage migration inhibitory factor (MIF) by ABIs and engagement of CD74, CXCR4 and CD44 receptors on CD206[hi] macrophages, could be important in maintenance of ABIs. MIF sustains pro-inflammatory immune responses by inhibiting p53-dependent apoptosis of macrophages[35] and controlling the expression of NLRP3, an essential step for activation of the NLRP3 inflammasome and subsequent activation of the IL-1b family of cytokines[36,37]. Its level have been shown to be increased in IPF lungs[38]. This may be significant as IL1-β -mediated inflammation appears critical in accumulation of basaloid intermediates (or damage-associated transient progenitors, DATP) in murine models of alveolar injury[10].

Numerically, the immune cell landscape is equally populated with adaptive and innate immune cells but there are clear differences in the way immune cells interact spatially with ABIs, and how they are organised spatially at different stages of disease. Our mathematical analysis allow us to generate an interactome which can be organised by a hierarchy of interactions - direct contact [positive correlation in numbers, $g(r = 20) > 1$ and CCN FDR < 0.05 - CD206[mid] and CD206[hi] macrophages], short range interactions [positive correlation in numbers and $g(r = 20) > 1$ but CCN > 0.05 e.g., resident (CD103[+]) CD4 T cells] and potentially long range interactions (positive correlation in numbers only e.g., CD4 T cells, CD14[lo] monocytes, proliferating monocyte-macrophages, neutrophil and NK cells) (Figs. 3a, d, 4c). Thus, foci of ABI-CD206[hi] macrophages together with more spatially distant immune cells (e.g., NK and neutrophil) could drive the growth of ABI at the leading edge of disease. Studies showing resident CD103[+] CD4 T cells interacting with MHCII-expressing epithelial cells[39] and augmenting homing of neutrophils to epithelium are[40] particularly interesting in this context since they suggest possibilities around initiation of injury in early disease stage by the CD103[+]CD4 T cells and ABI_b-DC ADJs as trigger for formation of ABIs.

Another noteworthy mention is the abundance of CD14[lo] monocytes, its correlation with that of ABI numbers, co-location with vascular endothelium in Early and Intm disease stages. By their markers, CD14[lo] monocytes are likely patrolling (or non-classical) monocytes (CX3CR1[hi]CD14[lo]CD16[+] in humans)[16,41,42]. Their co-location with vessels in our tissue further supports this. Resting, patrolling monocytes actively scan the vasculature. Inflamed or injured endothelium releases further chemoattractant to attract these monocytes[16]. Presence of CD14[lo] monocytes with blood vessels is indicative of inflamed vasculature in IPF, and supports an involvement, or association, between aberrant vascular health and aberrant alveolar regeneration.

Our main limitations are the following. We were unable to detect ATI cells, in main because at the time of study, we did not have specific ATI protein markers. Some ATI cells could be found within clusters of KRT5[neg-lo] ABIs (e.g., ABI_a) but the morphology of these cell types were examined on H&E sections (Fig. 2g) and were not compatible with ATI cell types. We believe they are probably found within the 27 UD cluster which formed 2.6% of epithelial cells. We focused on breadth of immune cells rather than depth and functional protein markers within the 33-plex panel so our immune cell profile lacked the deeper layer of markers that could indicate the subtypes of adaptive immune cells in particular. However as these cell types did not co-locate with alveolar epithelial cells. Finally our studies are descriptive and further work will be required to definitively determine the interactions observed and in silico mechanisms.

In conclusion, we present an unbiased spatial analysis pipeline that allows high resolution profiling of cells and identification of statistically supported spatial associations between cell types. A spatial interaction between CD206[hi] macrophages and ABI emerges as a major presence in the alveolar regenerating niche across all disease stages. This interaction could be a promising target to modulate abnormal stem cell differentiation, even in the most advanced stages of IPF.

## Methods

### Table of antibodies and reagents used in imaging mass cytometry and immunofluorescence

All antibodies, their catalogue numbers, final dilutions, and source are documented in Supplementary Table 5.

### Patients, samples, and ethical approvals

Diseased lung tissue was obtained from patients with a diagnosis of Idiopathic pulmonary Fibrosis (IPF) based on standard diagnostic criteria[43] and multi-disciplinary team (MDT) consensus, at the point of lung transplantation at the Institute of Transplantation, Newcastle Upon Tyne Hospitals NHS Foundation Trust. All patients provided written informed consent for use of their tissue via the Cellular and Molecular Mechanisms in Chronic Lung Diseases (EXPLANT) study which was approved by the NHS Research Ethics Service (11/NE/0291) and was sponsored by Newcastle Upon Tyne Hospitals NHS Foundation Trust (R&D ref 5885). We have considered sex/gender balance in selection of samples. There are 2 females and 7 males in our cohort, reflecting the demographics of the disease (male dominated disease). Gender was self-reported. Patients and relatives were not financially compensated. Number of patients and samples used was justified as follows - explanted lung tissue represents a highly specialised and rare resource. In the context of first in field work for IPF, no prior effect size estimates exist to inform a formal power calculation. The number of samples was determined by the availability of high-quality explanted lung tissue, and our previous experience in handling similar single cell data in COVID lung biopsies[15].

Formal histopathological analysis showed an overall progressive increase in the amount of collagen, smooth muscle and a reduction in the numbers of fibroblastic foci as disease progressed from Early to Intm and Adv stages (Fig. 1l, and Supplementary Table 2). In keeping with histopathology features of 'usual interstitial pneumonia' (UIP) in the IPF disease[43], lung sections displayed temporal and geographic heterogeneity in the fibrotic and inflammatory features across the Early, Intm and Adv regions. Given this, we subsequently treated each ROI as a single independent entity within each disease stage in our analysis.

Sections were stained with a 33-plex metal-tagged antibody panel (Fig. 1n, and Supplementary Table 3), which we designed to capture the breadth of the immune cell profile, and to identify regenerating alveolar epithelium, including the 'ABIs' and basal cells described by

Adams, Habermann and Kathiriya et al.[8,9,11]. Single cell segmentation, dimensionality reduction, cell clustering and annotation were performed according to the SpOOx 2.0 (Spatial Omics Oxford) pipeline, an extension of the SpOOx pipeline which we developed previously[15] (Fig. 1f–k and Methods). A total lung area of 61.89mm² was ablated and 313,818 single cells were obtained.

Disease-free control tissue was obtained from healthy donor lungs which were not used in transplantation and where the donor's next of kin had provided written consent for NHS REC approved research on the tissue (REC 16/NE/0230). Sections were taken again from the upper lobe as a direct comparison to the IPF tissue samples. Further samples from non-cancerous areas of lung cancer patients from the Oxford Biobank were also obtained under the NHS REC Number 14/SC/1060 for H&E exemplar staining.

## Slide preparation
Tissue sections for each donor were cut serially, at an IMC optimum thickness of 8 μm and mounted onto SuperFrost Plus™ Adhesion slides (Epredia). The primary slide from each serial deck was subject to H + E staining, followed by dual IHC staining of the following serial slide, with KI67 and PROSPC using the Discovery Ultra Research Staining System (Roche). Staining was performed by Novopath Research Service (NovoPath, Department of Pathology, Newcastle Hospitals NHS Foundation Trust, Newcastle upon Tyne, UK). Stained slides were scanned using the Aperio A2 slide scanner (Leica Biosystems). In combination, both H&E and IHC stains were used to navigate to areas of epithelial damage and repair in order to aid ROI selection. Slides were viewed and annotated in QuPath[44].

## Antibody validation and metal labelling
A 33-plex antibody panel was created to target the immune, structural and functional components of the IPF lung samples, and to identify the regenerating alveolar epithelial niche (Supplementary Table 3). Antibody clones were pre-validated as described in Hunter et al.[45]. In brief, validation and optimisation was carried out using two-colour immunefluorescence staining of appropriate positive control and lung tissue. Here, slides were imaged using a Axio Scan Z1 fluroscence microscope (Zeiss). Using Zeiss software, the microscope was configured for AF405, AF488, AF568, AF647, and AF750, and the entire area of the tissue section is selected using the software for higher resolution scanning, utilising a plan apochromat 40×0.95 korr M27 objective. Images were saved on a computer for further processing using custom Fiji/Image J macros[46,47].

Following verification of staining pattern and performance quality, approved antibodies were subject to lanthanide metal conjugation using a Maxpar X8 metal conjugation kit following manufacturer's protocol (Standard Biotools). Successful metal conjugation was verified by binding the antibody to iridium labelled antibody capture beads AbC™ Total Antibody Compensation Beads (Thermo Fisher) and acquiring on a Helios system (Standard Bio-tools).

See also Supplementary Information, 'Expanded information on tissue harvest, preparation and analysis of signal expression' and "Extended quality control (QC) steps"

## IMC staining
Sections for IMC analysis were baked at 60 °C for 2 h, deparaffinised in two changes of Histological grade xylene (Merck), followed by rehydration in decreasing grades of ethanol; 100%, 90%, 75% and 50%. Samples were washed twice in MiliQ ultrapure water and then submerged in Tris-EDTA 0.05% Tween 20, pH 9.0. Heat induced epitope retrieval was performed in the microwave for 20 mins. Slides were cooled to 70 °C, washed twice in MiliQ water followed by two washes in PBS (Gibco). Blocking of non-specific binding sites was performed using 3% BSA solution in PBS for 45 mins at room temperature. Sections were then subject to

staining with a cocktail of 2 DNA probe and 31 conjugated antibodies diluted in PBS containing 0.5% BSA and left to incubate overnight. Slides were washed in 0.2% Triton X-100 (ThermoFisher), followed by two washes in PBS. Nuclei were stained with Cell-ID™ Intercalator-Ir (Standard Biotools) for 30 mins at room temperature. Slides were washed in MiliQ water and left to air dry before IMC analysis.

See also Supplementary Information, 'Expanded information on tissue harvest, preparation and analysis of signal expression' and "Extended quality control (QC) steps"

## Hyperion (IMC) set up, quality control (QC) and sample acquisition
ROIs were ablated using the Hyperion Imaging System using CyTOF7 Software v7.0 (Standard BioTools) as described in Hunter et al.[45]. Sample data was acquired over 4 balanced batches. Prior to each sample acquisition, the Hyperion Tissue Imager was calibrated and rigorously quality controlled to achieve reproducible sensitivity based on the detection of ¹⁷⁵Lutetium. Ablations were performed at 200 Hz laser frequency to create a resultant MCD file per sample containing all data from ROIs. Raw MCD files were opened in MCD Viewer (Standard BioTools) where visual QC of the staining intensity and pattern was performed. All images were exported as 16-bit single multi-level TIFFs using the "export" function.

See also Supplementary Information, 'Expanded information on tissue harvest, preparation and analysis of signal expression' and "Extended quality control (QC) steps"

## Immunofluorescence
Paraffin-embedded human lung tissue sections were deparaffinized and each section was pre-treated using heat-mediated antigen epitope retrieval with sodium citrate buffer (pH 6) for 20 minutes. Then sections were blocked in 10% normal goat serum (Thermo Fischer Scientific, 50062Z) for 20 minutes and then incubated with Alexa Fluor® 488 Anti-Cytokeratin 17 antibody 1:200 dilution (Abcam, AB185032), CD15 antibody 1:200 dilution (Cell signalling Technology, 4744S), Pro-Surfactant Protein C Antibody 1:400 dilution (Abcam AB90716), Alexa Fluor® 647 Anti-Cytokeratin 5 antibody 1:200 dilution (AB193895), Anti-MRC1 Antibody 1:500 dilution (Atlas antibodies, AMAB90746), Anti-BDCA-2 antibody 1:100 dilution (Merck Millipore, MABF94), Alexa Fluor® 647 Anti-Fibronectin antibody 1:200 dilution (Abcam, AB198934) overnight at 4 ˚C. Each section is washed three times in TBS-T (0.1% Tween) and stained with Alexa Fluor 750 conjugated Goat anti Rabbit IgG or Alexa Fluor 488 or 568 conjugated goat anti-mouse IgM secondary antibody or Alexa Fluor 568 conjugated goat anti-mouse IgG1 for 30 minutes and washed three times in TBS-T (0.1% Tween) and mounted with Prolong platinum antifade Mountant with DAPI (Fischer Scientific) and the section slides were imaged using a Axio Scan Z1 fluroscence microscope (Zeiss).

See also Supplementary Information, 'Expanded information on tissue harvest, preparation and analysis of signal expression' and "Extended quality control (QC) steps"

## Data analysis
**Software and algorithms.** We utilised the SpOOx pipeline for data processing and analysis[15] The SpOOx pipeline incorporates Python and R based command line tools implemented as a semi-automated pipeline within a Ruffus framework. This pipeline provides end to end processing of imaging mass cytometry data encompassing processing of.TIFF files, segmentation and cell mask generation, extraction of signal intensity, dimensionality reduction, clustering and spatial analysis. For this work we improved the prior version of the SpOOx pipeline to include additional functionality for neighbourhood correlation function analysis and generation of statistically robust adjacency cell networks.

**Processing of MCD to.TIFF files.** MCD files were checked for staining and data acquisition quality using MCD viewer (provided by Standard BioTools). Following this initial QC assessment, images were converted to OME-TIFF format for segmentation.

**Segmentation and mask generation.** Cell segmentation was performed with the Mesmer library in DeepCell[48]. Nuclear markers DNA1 and DNA3 and cytoplasmic markers (CD45, CD68, CD206, HLA DR, CD14, CD15, CD11b, BDCA2, CD1c, CD11c, CD103, CD20, CD3, CD8a, CD4, CD69, CD56, EPCAM, KRT17, KRT5, CD31, aSMA, ProSP-C, CD127, Amphireg, CD161) were extracted and Z projected to generate single channel nuclear images and cytoplasmic TIFF images. These were contrast adjusted and processed with the Mesmer library reference as nuclear and cytoplasmic channels to generate a final single cell segmentation mask.

**Extraction of signal intensities.** Mean arcsinh transformed marker signal intensities were extracted from the segmentation masks with no filter for cell size area.

**Dimensionality reduction and clustering.** Clustering was performed with R phenograph[49] with parameter k = 15 on data following integration for condition (IPF) with Harmony for correction of sample and batch effects. Markers used for clustering are shown in Supplementary Table 3.

**Annotation workflow.** Cell phenotype annotation was performed using a sequential approach as described in Weeratunga et al.[15]. Briefly, clusters were initially defined using i) heatmaps showing median marker expression, ii) scaled expression density histograms and iii) cluster distribution plots. Based on this analysis we excluded clusters with uniformly low/negative marker expression and clusters found in only one sample. We termed a cluster undefined (UD) if they expressed markers which were found in more than two cell types e.g., variably, epithelial, several immune cell type and endothelial markers. We then examined the spatial location of these annotated clusters using cell centroid maps mapped on to the corresponding segmentation mask and an adjacent H and E region of interest. Some clusters demonstrated markers expressed uniquely on 2 different cell types and these markers showed adjacent expression in IMC images and IF. These clusters were denoted with 'ADJ'[15]. 'Adjacent cells' ('ADJ' in annotation) is a feature in tissue single cell mass cytometry that is not seen in suspension single cell flow cytometry and CYTOF methods, and single cell RNA sequencing, as these latter methods do not have the physically fixed locations as in embedded tissues. Thus, tissue analysis uniquely provides the spatial association of cells with each other. Since we do not have separation of single cells as we do in droplet single cell RNA sequencing or suspension single cell CYTOF or FACS, we have to computationally segment the cells in situ. See also Supplementary Information, 'Expanded information on tissue harvest, preparation and analysis of signal expression'.

For clustering stability, we use a biologically-driven approach to clustering and iterative loops of annotation (described in Supplementary Fig. 1) and once this is stable biologically, we check technical stability of this clustering using the method of quantifying cluster stability in IMC proposed by Bodenmiller in the The CATALYST R/Bioconductor package, (https://bodenmillergroup.github.io/IMCDataAnalysis/spillover-correction.html) (the 'silhouette score' and the 'neighbourhood purity score'). Silhouette score and neighbourhood purity score for clustering was similar to that proposed (average SCE 0.04 and average neighbourhood score 0.88).

**Analysis of differential cell abundance.** Differential cell abundance was examined between the defined states of IPF (early, intermediate and advanced) using the glmFit and glmLRT functions from the diffcyt package[50]. To account for differences in area of the analysed ROIs, area was used as a normalising factor.

## Spatial analysis

The spatial analysis pipeline in this manuscript follows[15], with the addition of Neighbour Correlation Functions (NCF)[20]. Here we give a brief technical overview of the components of this pipeline described in ref. 15 (the Quadrat Correlation Matrix, cross-Pair Correlation Function, Adjacency Cell Network and Topographical Correlation Matrix), before a more detailed description of the NCF.

**Quadrat correlation matrix.** The QCM follows methods described by Morueta−Holme et al. for inferring species associations from co−occurrence data[51], and provides a high-level overview of whether counts of each pair of cell types are observed to be correlated at short-range (a square of edge length 100 microns) more or less frequently than would be anticipated. Briefly, the QCM uses random shuffling of cell labels to generate a null distribution reflecting the correlation coefficients between each pair of cell types under random labelling. Observed correlation coefficients are compared against this distribution, to identify whether correlations occur by chance due to the underlying spatial structure of the points. The QCM produces $p$ values for the significance of the correlation, which account for multiple comparisons between a large number of different cell types. See Morueta Home[51] and Weeratunga[15] for further details.

**Cross pair correlation function.** Cross pair correlation function was performed as described previously[15]. Briefly, the cross pair correlation function (PCF), $g(r)$, is a spatial statistic which describes the correlation between pairs of points with specified labels A and B[20]. It describes this correlation at multiple length scales according to the variable $r$, with $g(r) > 1$ indicating that more pairs of points with label A and label B are observed separated by distance $r$ than would be expected under an assumption that the points are distributed by complete spatial randomness. Similarly, $g(r) < 1$ indicates that cell pairs are observed less frequently than expected at this distance. For full details of the PCF and its mathematical definition, see ref. 15 or[20].

**Topographical Correlation Map (TCM).** TCM was performed as described in ref. 15. The TCM is an example of a local indicator of spatial association (LISA)[20], and overlays a topographic "map" onto the ROI to show the local strength of positive or negative spatial association between cells of types A and B across the ROI[15,20]. It can be considered to represent the contribution of each cell of type A to the PCF for values of $r$ up to a given threshold, here 100 µm, or equivalently as a spatially resolved representation of contributions to Ripley's K function at $K(r)$[52].

**Neighbourhood correlation function.** The Neighbourhood Correlation Function (NCF) extends the standard pair correlation function to identify correlation between three (or more) classes of cell simultaneously[20]. We consider a point pattern in a finite 2D domain $\Omega$ such that each point $i$ has coordinates $x_i = (x_i, y_i)$. Suppose that each point is associated with a categorical label $A$, $B$ or $C$ and there are $N_A$, $N_B$ and $N_C$ points with labels $A$, $B$ and $C$ respectively. Then there are $N_A \times N_B \times N_C$ possible ways to select a set of three points such that we have one point with each label. For each such set of three points $\zeta$, we calculate the smallest circle which can enclose all three points, which we call a "neighbourhood" and define to have radius $R(\zeta)$.

The NCF at radius $r$ compares the total number of neighbourhoods of radius $r$ against the number of neighbourhoods of size $r$ that would be expected if points were distributed according to complete spatial randomness (CSR). Thus, the NCF for points with labels $A$, $B$ and

$C$ is defined as:

$$NCF_{ABC}(r) = \sum_{a=1}^{N_A} \sum_{b=1}^{N_B} \sum_{c=1}^{N_C} \frac{I_r(R(\{x_a, x_b, x_c\}))}{N_A \times N_B \times N_C \times p_3(r)} \quad (1)$$

where $I_r(R) = 1$ if $R \in [r, r+dr)$, $dr > 0$, and $p_3(r)$ is the probability of three points chosen under CSR being found within a neighbourhood of radius $r$. $p_3(r)$ is defined within $\Omega$ as $p_3(r) = \lim_{M \to \infty} \frac{\sum_{i=1}^{M} I_r(R(\{x_i^1, x_i^2, x_i^3\}))}{M}$, where $x_i^1$, $x_i^2$ and $x_i^3$ are locations sampled under CSR within $\Omega$. Practically, $p_3(r)$ is approximated by taking large $M$ (here $M = 10^7$). Values of the NCF greater than 1 are consistent with clustering of the three cell types together at distance $r$, while values less than 1 are consistent with exclusion of one or more cell types from the others at the length scale $r$. For more details, please see ref. 20.

The resulting radii were the binned into 5 μm intervals to produce a histogram. Bootstrapping was performed by randomly shuffling the location of the three cell types and calculating the appropriate histogram. The average and 0.05 and 0.95 percentiles for each interval from the 1000 bootstrapped histograms were used for comparison with the observed value.

To calculate NCFs for a condition, comprising several ROIs, the radii for all ROIs in a condition were combined and the histogram recalculated. For bootstrapping, the bootstrap histograms for each ROI in a condition were combined resulting in 1000 histograms. Again, the average and 0.05 and 0.95 percentiles for each interval from these histograms were used for comparison with the observed value.

**Contact cell networks**. Adjacency cell networks were calculated as previously described[15]—for each co-locating cell pair A/B [g(r = 20) > 1], the total number of B cells in contact with A cells was calculated. For the contact cell network (CCN), we ask if contacts were significantly different. Z -scores were calculated by randomly shuffling cell labels and recalculating contacts[53]. The average and standard deviation (sd) of a thousand bootstrap values were the used to calculate z-scores.

$$z_{AB} = (O_{AB} - \mu(N_{AB}))/\sigma(N_{AB}) \quad (2)$$

where:
- $z_{AB}$ is the z-score between cell types A and B
- $O_{AB}$ is the number of observed contacts
- $(\mu N_{AB})$ and $\sigma(N_{AB})$ represent the mean and standard deviation of the number of contacts in the bootstrapped data

*P* values were calculated using the *norm.sf* function from *scipy* package[54] and corrected for multiple comparisons with *multitest* from the *statsmodel* package[55]:-

*from scipy.stats import norm*
*from statsmodel.stats import multitest*
*p values = norm.sf(np.asbsolute(z_scores)) *2*
*fdr_corrected p values = multitest.fdrcorrection(pvals)*

To obtain the z-score for an A/B interaction in a condition, comprising several ROIs, a weighted average was used, based on the number of A Cells in in the interaction. This was calculated by the sum of

*A/B zscore * A count / sum of A counts*

*P* values for each condition were calculated combining the individual *P* value for each ROI using the *stouffer* method and weighted by the A counts. This was implemented by the *combine_P value* method in the in *scipy* package[54]

*from scipy.stats import combine_pvalue*
*combine_pvalues(pvalues,"stouffer", Acounts)*

Again, the p-values were corrected using the *multitest* function from the *statsmodel* package (see above)

## Single cell transcriptomic interrogation
Within the single cell data from Habermann et al.[9] macrophage, monocytes, cDCs and pDCs from control and IPF patients were further sub clustered using Seurat R package (ver 4.3.0)[56]. Highly variable genes were identified by fitting the mean variance relationship for each sample to avoid sample-to-sample variation and to retain variance within the samples. We then performed principal-component analysis using top 2,000 highly variable genes. Scree plots and Jackstraw permutation tests were used to determine significant principal components (with P cut-off < 0.01) in the data. Subsequently, we performed batch correction using Harmony (ver 0.1.1)[57]. A k-nearest neighbour (kNN) graph was constructed from cells in the reduced dimension space in Seurat based on the top 20 batch-corrected components. Cells were then clustered in this on this neighborhood graph using the Leiden algorithm for modularity optimisation where the resolution parameter was set to 0.7.

Based on cluster marker gene expression Plasmacytoid dendritic cells (pDC), classical DCs (cDC), monocytes and macrophages were identified, and the monocyte and macrophage clusters re-clustered as a single entity to identify the CD206hi macrophage subset. We then merged these with annotated KRT5-/KRT17+ cells and transitional AT-II for downstream receptor ligand analysis.

Potential ligand receptor (LR) interaction between different cell-types was analysed using CellChatDB statistical analysis (ver 1.6.1) with built in human ligand-receptor database[21]. Potential ligand–receptor pairs, cell–cell interactions were identified based on mass action models, along with differential expression analysis and statistical tests on cell groups. We used the standard workflow; Preprocessing functions, identifyOverExpressedGenes and identifyOverExpressedInteractions with the default parameters were applied followed by communication probability computation between interacting cell groups with tri-mean set at 0.1. Subsequently we ran the core functions computeCommunProbPathway and aggregateNet using the standard parameters. Finally, the functions, netVisual_bubble were applied to the network to determine the senders and receivers and to generate a LR interaction plot.

## Reporting summary
Further information on research design is available in the Nature Portfolio Reporting Summary linked to this article.

## Data availability
The spatial mass cytometry data files (MCD) generated in this study have been deposited in the Zenodo database under accession code 10930946[58]. The processed source data used in the analyses presented in the paper is available for download as an online resource in Multi-Dimensional Viewer (MDV) accessible with the hyperlink, (https://mdv.molbiol.ox.ac.uk/projects/mdv_project/7430). Specific source data for graphs are also provided in the source data file. Source data are provided with this paper.

## Code availability
The complete code for the Spatial Omics Oxford pipeline (SpOOx v2.0) is available as a GitHub repository under the GPL license (https://github.com/LingPeiHo/Ho_Taylor_Byrne-SpOOx-2.0) (https://doi.org/10.5281/zenodo.15719295)[59]. The Multi-Dimensional Viewer code is available under the GPL license: (https://github.com/Taylor-CCB-Group/MDV), (https://doi.org/10.5281/zenodo.8324918)[60].
**Software source and identifier:**
Software used in data processing and analysis in this paper is presented in Supplementary Table 6.

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

## Acknowledgements

The study is funded by combination of grants from MRC UKRMP Grant MR/S020918/1; NIHR Oxford Biomedical Research Centre grant; University of Oxford Medical Science Division and Newcastle University Flow Core Facility. LPH is supported by MRC (grant CFR01480) and the NIHR Oxford Biomedical Research Centre. AJF is supported by the National Institute for Health and Care Research (NIHR) Blood and Transplant Research Unit in Organ Donation and Transplantation (NIHR203332), a partnership between NHS Blood and Transplant, University of Cambridge and Newcastle University. The views expressed are those of the author(s) and not necessarily those of the NIHR, NHS Blood and Transplant or the Department of Health and Social Care.

## Author contributions

P.W. led the conceptual and intellectual development of the SpOOx V2.0 pipeline with L.P.H., and ran all analyses, writing, interpretation of data, data organization, figures, tables, and writing of the paper with L.P.H. P.W. and M.S. performed the MDV analyses and computational software and pipeline extension. M.S. generated all computational output utilizing the SpOOx 2.0 pipeline and was involved in data organization and visualization on MDV in conjunction with P.W., J.B., and H.B. J.B. designed the mathematical algorithms, tested these in silico, applied these to study and critically appraised interpretation of mathematical tools with P.W., H.B and L.P.H. H.B. led mathematical development, interpretation and provided critical analyses of overall final synthesis of data. C.C. performed all the expert histopathology analyses in discussions with L.P.H. and A.F. L.D. designed the IMC panel with L.P.H., and contributed to generating and analyses of the MCD images during analysis of overall data and critical review of data interpretation and methods text. C.V. performed all optimisation of immunofluorescence (IF) validation staining, stained all samples and acquired further control samples for IF staining. T.H. analysed and contributed to annotation the aberrant alveolar epithelial regeneration, and ran transcriptomic analysis to inform annotation and matching of ABIs to prevailing literature. J.M.W., A.A. and P.W. performed the single cell transcriptomic analysis including CellChat analyses and generated graphical output. J.S. was involved in conception of study, acquisition of parts of funding of study and review of results. A Filby and B.H. optimised and performed the I.M.C. staining and generation of M.C.D. files. AFisher was involved in conception of study, acquisition of parts of funding of study, sample acquisition, ethics application, governance of sample harvesting, review of results and writing. L.B. leads the Newcastle lung sampling programme, and ensured governance of sample harvesting with AFisher. R.B. collected explanted lungs and processed lung tissue samples. L.P.H. conceived and led the study and the conceptual and intellectual development of SpOOx v2.0 pipeline with P.W., J.B. and H.B., analysed all aspects of the data and interpreted all the output with P.W., wrote the paper with P.W., performed extended analyses of histopathology with C.C., corroborated all IMC and IF staining with P.W. and C.V., interpreted all mathematical applications and output with P.W., J.B. and H.B., synthesised and amalgamated overall output, and obtained funding for all parts of the project. All authors read the paper and were involved in corrections and refining the final paper.

## Competing interests

The authors declare no competing interests apart from JS who is part of the Scientific Advisory Board for Mogrify (fees to Newcastle University).
