## [Transparent Peer Review file · Nature Communications]

Temporo-spatial cellular atlas of the regenerating alveolar niche in idiopathic pulmonary fibrosis

Corresponding Author: Professor Ling-Pei Ho

Version 0:

Reviewer comments:

Reviewer #1

(Remarks to the Author)

The authors have aimed to understand the spatial landscape of IDF by employing IMC and a plethora of spatial tools. The work is promising but I do have a few comments that I would like to see addressed. As my expertise lies with IMC and not with the disease in question I have focused my review on the methodological aspects.

Major comments

1. I would urge the authors to be careful with the distinction between low, intermediate and high expression of a marker when using FFPE tissue sections, for instance the KRT5^{low/mid} and KRT5^{hi}. Intensity differences in FFPE IMC are extremely influenced by technical factors outside the scientists control such as fixation time, time between operation and fixation, tissue size, tissue viability. But also analysis factors we do not encounter when analysing single cell data such as: the size of cells, the size of the nuclei, where they are cut, etc. Therefore, it is difficult to say if an intensity difference is biological or technical.

2. The authors describe that very little CD14^{high} monocytes are found, and mainly adjacent to DC. Supplemental figure 4B does show a very high abundance of CD14⁺ cells though. Is this just in this one image? It would be useful if the authors highlight if those cells are the CD14^{lo} monocytes or macrophages, for instance by adding CD68 to the IMC image.

Furthermore, if these are classified as CD14^{lo} monocytes, I do not see a difference in the CD14 intensity between highlighted cells and the others. Thus, I am not convinced that there is a difference and the authors should provide an example of these differences between CD14^{high} and CD14^{lo} cells within the same image.

3. Figure 3B shows the colocalization of different celltypes, it would be of value to show how the highlighted neighbourhoods differ between samples, for instance in a simple boxplot depicting the abundances of the neighbourhoods. If only for the neighbourhoods that are highlighted.

4. The authors investigate the spatial localisation in many ways which can give a lot of different information but also can be difficult for the reader to follow. For instance, why do the authors investigate the colocalization of CD103⁺CD4s, ABIs and CD206 mac in figure 3g in a very different way than the general community analysis in figure 4. What does the use of the different methods add?

5. In figure 5, the authors investigate the occurrence of direct cell-cell neighbourhoods, instead of a 20 μ m radius in figure 3. This is an important question, however, although it is a bit difficult to compare due to the different visualisation, I am not convinced that there is much difference between the results of these two, as 20 μ m is already very close. Indeed, the same interactions are found. Can the authors highlight what this adds and why is this analysed with a completely different approach rather than using the approach of figure 3 and minimising the radius.

6. "The pattern of spatial co-location changed at the leading edge of disease (Intm), where CD206^{hi} macrophages appear to be the central connector for the ABI and immune cell communities (Fig 5E), linking the entire ABI network to the immune network". This sentence, and the visualization in 5d-f are a bit strong from what can be taken from the data. The data merely shows that interactions between cell 1 and 2, cell 2 and 3, cell 3 and 4 etc occur, not that there is one mega ABI-immune network.

Minor comments

1. Affiliation 6 is not linked to an author
2. temporal-spatial niche instead of temporo
3. throughout the text, mm is used instead of μm
4. co-localised instead of co-located

(Remarks on code availability)

Reviewer #2

(Remarks to the Author)

Weeratunga and colleagues, used a 33-plex single cell imaging mass cytometry to characterize cell populations in lung samples from 7 patients with end stage IPF (obtained at lung transplantation) and 2 controls (donors), focusing on epithelial cells and immune cells. They assessed 3 areas per patient. Then, they reanalyzed publicly available scRNAseq data to determine whether ligands-receptors could be identified.

1-I do not understand why the authors suggest that alveolar metaplasia is a feature of alveolar regeneration. I would suggest instead to be purely descriptive in the results section and call this alveolar metaplasia and discuss later whether this is alveolar regeneration.

2-A major issue in this study is the way the authors analyzed the samples. They identified samples as Early/End stage/intermediate on the basis of morphology. However, when looking at figure 1I, it appears that what the authors call "Early" is already very severe disease since one cannot recognize any normal alveolar structure. This is probably why the authors did not find a difference in the overall proportion, distribution, or abundance of immune and epithelial cells in the three categories of lesions. This is a major flaw to this study.

3-The authors identified a population of epithelial cells with low expression of CD11c and CD45. This looks like an artifact where myeloid cells in close proximity to epithelial cells are considered as "positive" by the system. What is the evidence for the actual existence of these cells? Do they identify these cells also scRNAseq data bases?

4-The authors observed correlations between ABI and immune cell infiltration, with a statistical link suggesting colocalization between ABI and CD206mid macrophages, CD206hi macrophages and CD103+ CD4 T cells. More specifically, they suggest that CD206hi macrophages were in close contact with ABI_b-DC ADJ and CD206mid macrophages with ABI_a cells. Their in silico analysis suggest that fibronectin and MIF are potential ligands in this interaction. They should provide experimental data for a communication between macrophages and epithelial cells, through these ligands, driving the differentiation of epithelial cells.

(Remarks on code availability)

Reviewer #3

(Remarks to the Author)

Weeratunga and colleagues take a single-cell spatial proteomic approach to explore the temporo-spatial cell atlas of the regenerating alveolar niches. Their extensive efforts have produced a high-resolution map from well-characterized and registered idiopathic pulmonary fibrosis (IPF) samples from different disease stages. By conducting comprehensive analyses and thorough comparisons, the authors have identified CD206hi alveolar macrophages as central immune cells in the immune-alveolar epithelial interactome.

While I have a generally positive view of the manuscript, several issues should be addressed. The reproducibility of the single-cell data analysis across donors/conditions needs further attention, additional assessments are required to evaluate the robustness of the cell clustering and annotation, and there are missed opportunities for more thoroughly characterizing receptor-ligand cell communities in spatial space. Furthermore, the authors should refine their statements and conclusions to ensure they are presented with greater precision and rigor. In addition, the following points must be addressed to justify publication in Nature Communications:

Major comments:

1. Please provide detailed quality control for each sample/experiment. For example, an average number of antibody signals per cell, any potential low-quality cells were removed, what criteria were used, and how the cutoffs were determined.
2. For each antibody marker expression, whether the distribution is consistent between samples and/or conditions. Any potential low-quality antibody markers?
3. I appreciated the detailed description of cell clustering steps. Please quantify whether the clusters are stable and robust. Whether the single or the combination of multiple markers used for annotation are specific. Any potential over-splitting or under-splitting? Is there any potential bias between batches and donors?
4. In Fig 1O, it would be nice to show 2D embedding. Whether certain cell populations be better separated by sub-clustering?
5. For cell cluster annotation, please also compare and confirm with previously published reference maps (e.g., scRNA-seq).

6. Whether the correlation in Fig 2k relatively consistent across donors/ROIs?
7. In the cross-PCF analysis, could authors clarify why epithelial cell types were assigned as 'anchor cells'? Given the different abundance of cell types, I am just curious if picking other cell types as 'anchor cells' will result in the same conclusion.
8. If I understand correctly, the ACN analysis was construed by the cross-PCF analysis. It was nice to see the consistent results in Fig. 4A-B (epithelial cells as the anchors), but could the authors explain what additional info we can learn from the ACN results? Is the selected radius parameter ($r=20$) correspond to the distance between any two cells?
9. Can authors explore the correlation of 33 makers between RNA expression level and antibody level? Is it possible to project scRNA-seq data to the spatial space? This is a missed opportunity to check if the receptor-ligand cell communities predicted by CellChat have a higher possibility of physical contact between various cells in the spatial space.

Minor comments:

1. Does the "ROI" represent the region of interest?
2. The figure panels are labeled in lowercase in the figure but uppercase in the main text.
3. Fig 2K is actually a table and text in small size, maybe change it to bar plots?
4. The text size in Fig 3c is unreadable.
5. Fig. 4C-D were not cited in the main text.

(Remarks on code availability)

I have checked the code, the description of the methods and instruction were detailed.
The scripts for this paper's datasets may need to be provided.

Version 1:

Reviewer comments:

Reviewer #1

(Remarks to the Author)

I would like to thank the authors for their thorough response and my questions have been answered sufficiently.

(Remarks on code availability)

Reviewer #3

(Remarks to the Author)

Overall, I greatly appreciate authors detailed explanation, extensive analyses, and attempt to align with scRNA-seq and test the receptor ligand communication in an independent dataset. The authors had done substantial work to address most of my concerns by thoroughly benchmarking, revising the manuscript considerably, and including detailed descriptions. The other reviewers of this manuscript had some insightful questions that the authors have also addressed. I enjoyed reading this round of revision. However, I would recommend the authors make additional test and/or modeling to make sure those immune cell populations are robustly identified. A few concerns are detailed as follows:

1. I appreciated the detailed explanation of quality control steps.
2. Sorry about the potential misleading. Could you test if the antibody marker expressions are consistent across samples for each condition, say the early stage. In other words, do you see any potential bias from one sample, a field of view, which should be excluded for downstream analyses?
3. First of all, I don't believe the fewer number of features (31 antibodies) is an issue for robustness testing. If the selected features are truly informative and of good quality, only two features can still separate clusters well. Second, while I understand that the overall silhouette score and neighborhood purity score can achieve the recommended cutoffs on average, certain cell clusters might still face potential over-, under-splitting or could be of lower quality. For example, cluster #24 has a lower silhouette score but a higher neighborhood purity score—should I interpret this as a low-quality cluster that potentially contains a mixture of cells from multiple cell types? Third, the cell type annotation for each cell cluster in Fig. R15&16 is not very clear to me. The authors mentioned 29 clusters in the main text but showed 35 cell clusters in the rebuttal letter, which suggests they might have manually merged some clusters. If that's the case, consider training a simple classification model (e.g., random forest, SVM) to determine if the captured signals are of high quality and can be used to distinguish the 29 cell clusters effectively. These results would provide strong supporting evidence for the cell annotation based on prior knowledge. Lastly, I totally understand that it's not easy to annotate immune cell population, but the authors should use a reasonable cell cluster resolution that is best supported by the captured signals.

(Remarks on code availability)

Reviewer #4

(Remarks to the Author)

This revised manuscript presents a spatial analysis of the regenerating alveolar niche in idiopathic pulmonary fibrosis (IPF) using Imaging Mass Cytometry (IMC) and an advanced spatial analysis pipeline. The authors responded well to the comments, however, a few open questions need to be resolved.

Comment 2:

Thank you for the detailed clarification regarding the rationale for stage classification. While it is now clear that the terms “Early,” “Intermediate,” and “Advanced” are intended to represent relative stages within explanted, end-stage IPF lungs, the terminology may still be misleading to readers. To enhance clarity and avoid potential confusion, it is recommended to consider alternative descriptors such as “Less fibrotic,” “Moderately fibrotic,” and “Severely fibrotic,” or to include consistent reminders throughout the manuscript that these classifications refer to advanced-stage disease. Such adjustments would improve interpretability for a broader readership.

Comment 3:

Thank you for the clarification and the detailed explanation of the annotation process. While the rationale is technically sound, it may still be difficult for readers to interpret the ABI_b–DC ADJ cluster as a distinct entity without a clearer definition. In particular, Figure 2C shows that this cluster expresses CD45, and it remains unclear whether CD11c is also present. I recommend explicitly stating in the manuscript that this cluster represents a spatial designation—reflecting proximity between epithelial cells and immune cells—rather than a novel hybrid phenotype. Additionally, please provide further evidence and characterization of this cluster to help avoid potential misinterpretation. It would be helpful to clarify how spatial adjacency was operationally defined and whether segmentation overlap could have contributed to the apparent co-expression of markers typically restricted to immune cells.

Comment 4:

It is understandable that direct experimental validation of the predicted ligand–receptor interactions is beyond the scope of this study. However, the manuscript would benefit from using more cautious language when discussing these findings. Specifically, the roles of fibronectin and MIF should be described as predictions rather than confirmed mechanisms. This will help set appropriate expectations for readers and avoid overstating the biological significance of the computational results.

(Remarks on code availability)

Version 2:

Reviewer comments:

Reviewer #3

(Remarks to the Author)

I greatly appreciated the thorough response, additional analyses, and my questions have been well addressed.

(Remarks on code availability)

REVIEWER COMMENTS

Responses in blue

Reviewer #1 (Remarks to the Author):

The authors have aimed to understand the spatial landscape of IPF by employing IMC and a plethora of spatial tools. The work is promising but I do have a few comments that I would like to see addressed. As my expertise lies with IMC and not with the disease in question I have focused my review on the methodological aspects.

Thank you for reading our M/S so carefully. We very much appreciate your time, expertise and experience.

1. I would urge the authors to be careful with the distinction between low, intermediate and high expression of a marker when using FFPE tissue sections, for instance the KRT5^{low/mid} and KRT5^{hi}. Intensity differences in FFPE IMC are extremely influenced by technical factors outside the scientist's control such as fixation time, time between operation and fixation, tissue size, tissue viability. But also analysis factors we do not encounter when analysing single cell data such as: the size of cells, the size of the nuclei, where they are cut, etc. Therefore, it is difficult to say if an intensity difference is biological or technical.

Overall, we agree with cautious and circumspect 'calls' of low, mid and high expression of a marker in FFPE tissue sections. However, we are keen to make the following points, related to our optimisation process prior to final staining and examination of expression intensity.

Tissue factors. We have a very rigorous and well-established protocol designed to minimise variation in tissue harvesting and processing. Only one transplant physician and one lung pathologist are involved in tissue harvesting (Newcastle Lung Transplant Centre), there is a well established harvest, store and lung sectioning pipeline (Newcastle histopathology research pipeline), and a direct pathway from patient to bench, using standardised research SOP. Acquisition was done on one Hyperion machine sequentially over 2 weeks, in three balanced batches and all checks for technical stability of the machine were performed before (and during) each acquisition.

Each H&E stained, next-in-sequence lung section is reviewed by a senior pathologist (>30 years experience) and two senior pulmonologists (>20 years experience) to identify regions of interest (ROIs). Small ROIs were used (1x1mm), allowing us to choose areas that demonstrate excellent sectioning without folds and with consistent staining, enabling us to avoid areas with mucus impaction and high numbers of red blood cells which can affect staining.

Optimal thickness of lung sections was pre-determined. Performing multiple rounds of staining using antibodies enabled us to select a tissue thickness that most likely corresponds to a single cell. For example, CD3 T cells are either CD4+CD8- (CD4 T cells) or CD4-CD8+ (CD8 T cells) in healthy tissue. An optimal lung thickness will capture these two cell types as distinct entities, while a section that is too thick may capture a false cell cluster of CD4+CD8+ cells if these two cell types are on top of each other (Figure R1).

These well established, tested and standardised pre-staining provisions and optimisation minimise the kind of problems that this reviewer has justifiably raised.

Figure R1. Schematic demonstrating the impact of tissue section thickness on marker detection and annotation in imaging mass cytometry

Analysis factors. Expression levels of antibodies is only one of several factors leading to the final decision on an annotation/identity of the cell clusters. Biological knowledge has a critical contribution, particularly in our setting. Thus, the use of low, mid and high expression is never undertaken in isolation; it is always used in conjunction with biologically expected change in a combination of markers. The calling of high and low expression is also restricted to proteins which are expected to vary for a biological reason, typically differentiation, plasticity or phenotypic subtyping. For example, monocytes are precursors of macrophages, and lose CD14 while gaining CD68 expression as they differentiate into macrophages. These reciprocal changes are observed in these cell types - CD14^{lo} monocytes have high CD68 expression while those with high CD14 have low CD68 expression. Thus, the calling/annotation of CD14^{lo} monocytes does not rely on CD14 expression alone but also on the concomitant and appropriate expression of other biologically related proteins, that have been well established biologically for that particular group of cells. This is also the case for KRT expression in aberrant basaloid intermediate cells where a collection of markers, KRT5, KRT17, pro-SpC, EPCAM and CD45 are evaluated together before a final call is made for the identity of a cell.

These data are found in Suppl Table 4 which shows expression of all relevant antibodies together for the biologically appropriate groups of cell types (eg differentiating monocyte-macrophage myeloid cells and differentiating alveolar epithelial cells). Please find Suppl Table 4 below for convenience (Figure R2):

Myeloid cell clusters and selected epithelial cells and their relative expression of relevant markers								
Name	Cluster number	CD11c	CD14	CD1c	HLADR	CD68	CD206	BDCA2
CD206 ^{hi} mac	1	+	++	+++	+++	+++	+++	+
CD14 ^{hi} mono-DC ADJ	9	++	+++	+++	+++	+/-	-	+
CD14 ^{lo} mono	17	-	++	+	++	-	-	-
Prolif mono-mac	18	-	++	+	++	-	-	+
CD206 ^{mid} mac	22	+	-	+	++	++	++	-
CD206 ^{neg} mac	29	-	-	-	-	+/-	-	-
ABI b-DC ADJ	10	+	++	+++	+++	+/-	+/-	+
Basal	13	-	++	+++	+/-	-	-	+
27 UD	27	-	++	+	+/**	-	-	+
ATII	6	-	++	+	+++	-	-	+

Supplementary Table 4. Annotation of clusters, and expanded notes on annotation.

Figure R2. The original Suppl Table 4 showing myeloid cell clusters and epithelial cells' relative expression of relevant markers.

Of course, the use of low, mid and high expression is done after consolidation of numerical rather than visually determined signals, aggregated from the average expression for all cells in any one cluster, in order to reduce the contribution of individual/single poor or erroneous staining. Multiple iterative 'loops' of analyses were performed. In the following paragraphs, we summarise how we have done this, in order to address the comment '...also analysis factors we do not encounter when analysing single cell data such as: the size of cells, the size of the nuclei, where they are cut, etc..'

A computational analytical algorithm (Deep Cell)¹ first segments the cells, identifying the nucleus as the centre of a cell, and using the cytoplasmic and cell surface markers to map out the rest of each cell. Segmentation is performed as shown in Figure R3A. Edges of unusually shaped cells (see below) that are captured without nucleus are discarded computationally. The average intensity of each marker for each cell is then obtained and submitted for unbiased dimensionality reduction and clustering process with R – Phenograph. All samples are integrated and batch correction is performed (Harmony) and, finally, an overall representation of the marker expression per cluster across all samples (expression density histogram) is generated (Figure R3B-C). High numbers of single cells (n= 313,000) help to reduce the contribution of signals due to technical variability; and consistent, computationally applied rules about which cells to keep and which to discard across all cells and all samples provide a final, unbiased single cell landscape.

Figure R3. A. Diagram demonstrating the process of single cell segmentation by DeepCell B-C. UMAP showing final annotated clusters and scaled marker expression density plots for selected cell clusters

Once these steps have been completed, we proceed to annotation which is a very involved process, as demonstrated in Suppl Figure 1, which we reproduce here for convenience (Figure R4):

Figure R4. Original Suppl Figure 1 showing annotation steps

We have slightly modified the prose of the response above and placed it in Supplementary Material under the heading “Expanded information on tissue harvest, preparation and analysis of signal expression”.

2. The authors describe that very little CD14high monocytes are found, and mainly adjacent to DC. Supplemental figure 4B does show a very high abundance of CD14+ cells though. Is this just in this one image? It would be useful if the authors highlight if those cells are the CD14lo monocytes or macrophages, for instance by adding CD68 to the IMC image. Furthermore, if these are classified as CD14lo monocytes, I do not see a difference in the CD14 intensity between highlighted cells and the others. Thus, I am not convinced that there is a difference and the authors should provide an example of these differences between CD14high and CD14lo cells within the same image.

We appreciate how you have come to this conclusion. Our response below demonstrates the points made in our response to Comment 1 above.

The CD14^{hi} monocytes were not annotated on the basis of CD14 expression alone (each cell type is a composite of all the marker expression, with some biological ‘driver markers’, in this case CD14 and CD68, (see cluster 9; Suppl Fig 1 and interpretation in Suppl Table 4). These markers cannot be appreciated by eye alone on an image as the signal variation is often not visible and can also be subtle for any one signal in one cell in one sample. Rather, all the unmanipulated data signals representing expression in ALL cells in ALL samples are compiled together and expressed in a density expression histogram, as shown in Suppl Figure 1 (provided in Figure R3C for some clusters). From these, we interpret the combination of relevant or driver markers for each UMAP-determined cluster. This ensures no manipulation, a numerical determination of expression and unbiased global comparison.

The image that the reviewer highlighted in Suppl Figure 4 is a good example of how visual inspection alone is unable to determine high and low expression. Using the visualisation tool that we have developed (<https://mdv.molbiol.ox.ac.uk/>), these magenta CD14-expressing cells are in fact identified as CD14^{lo} monocytes and CD14-expressing proliferating mono-mac cells (Figure R5)

Figure R5. Original Suppl Fig 4B on left showing MCD file staining for CD1c and CD14 with corresponding centroid plots beneath it with the identity of these cells (CD14^{hi} mono-DC ADJ). New centroid plots on right of the same ROI identifying the other CD14+ cells as CD14^{lo} monocytes (Cluster 17, see Suppl Table 4) and proliferating mono-mac (Cluster 18). Both these cell types are shown together in the upper panel.

3. Figure 3B shows the colocalization of different cell types, it would be of value to show how the highlighted neighbourhoods differ between samples, for instance in a simple boxplot depicting the abundances of the neighbourhoods. If only for the neighbourhoods that are highlighted.

Thank you for this query. For clarification, there are no 'neighborhoods' as such in Figure 3B. Rather, this mathematical method (cross-PCF) identifies pairs of co-locating cells. Thus, the cell pairs highlighted in Figure 3B, e.g. CD103⁺ CD4 T cells and ABI_b-DC ADJ cell types, were found to be statistically co-located with each other compared to complete spatial randomness in the lung sections for Early disease.

We, therefore, assume you mean how these pairs of co-locating cells vary in their strength of co-location between samples. If so, please find below a violin plot showing how the strength of co-location varies between samples. We have now added this information (Figure R6) to the Suppl Data in the M/S. The values are normally distributed (D'Agostini and Pearson test) for ABI b-DC ADJ and CD206^{hi} mac pairs in Intm and Adv stages.

Figure R6. Violin plots showing the distribution of $g(r=20)$ in individual IPF samples for cell pairs identified as significantly co-located compared to complete spatial randomness in the given disease states – see also figure 3B in main manuscript.

In terms of 'neighbourhoods', we reserve this term for data that encompass more than one pair of cell types. For example, in Figure 4A-B, we examined all cell types (not just immune cells) found to be co-located with our cell type of interest [aberrant basal intermediates (ABIs)] in different stages of disease. In Figure 3G-J, three cell types derived using the NCF (neighbourhood correlation function) are shown. The error bar in Figure 3G shows how these neighbourhoods differ between samples, and corresponds to the 95% confidence interval derived by bootstrapping for each displayed $NCF_{c_1, c_2, c_3}(r=30)$, based on the number of minimum enclosing circles at the 5th and 95th centile. We have added this to the legend.

4. The authors investigate the spatial localisation in many ways which can give a lot of different information but also can be difficult for the reader to follow. For instance, why do the authors investigate the colocalization of CD103+CD4s, ABIs and CD206 mac in figure 3g in a very different way than the general community analysis in figure 4. What does the use of the different methods add?

Thank you for your query. The different methods add strength to each other and provide different information. Thus,

1. The pair-wise comparison provided by the cross-PCF (Fig 3B) identifies, in an unbiased way, the pairs of cell types that are spatially associated with each other more frequently than would occur by chance (i.e., under complete spatial randomness)

2. The triplet comparison provided by the NCF (Fig G-J) identifies, in an unbiased way, trios of cell types that are physically associated with each other
3. The neighbourhood analysis in Fig 4 brings together all pairs of cell types that are spatially co-located with our cell type of interest.

We often use the following analogy in our talks. Let's say we have many different animals in a large safari park and we are interested in which animals are found with the elephants. The cross-PCF identifies two types of animals that are found more than expected within a certain distance of each other (say 1 m radius). So, let's say that the cross-PCF identifies the elephant and the gazelle as being co-located within 1 m radius of each other more often than would be expected if they were randomly distributed within the 500 km² safari park, despite there being 100 other types of animals roaming around the park. The NCF identifies triples of animals (say elephant, lions and gazelles) that are found closer together than would be expected in the park. Neighbourhood cross-PCF analysis (Figure 4) then asks, for the elephants, apart from animals, what other structures are also found associated with elephants e.g. acacia trees, lakes.

The network analysis (ACN and CCN) asks, of the animals that are found close together, are they in direct contact with each other, not just found within a certain distance of each other. This is important as animals that are in direct contact could, let's say, pass on diseases to each other. Thus, in our sample, directly contacting cells could relay signals via a receptor-ligand interaction.

Where we examined how the cell types differ in their co-location in the three different disease locations/stages (Early, Mid and Advance in the lungs), the analogy could be that we ask if the elephants and gazelles found spatially associated with each other are found in parts of the safari park where there are young trees (analogous to mid disease stage in IPF) or where the trees are mature (Advanced) or where they also only seedlings (Early).

This method approaches the safari park without a specific, *a priori* hypothesis about whether we expect elephants to be closer to gazelles than expected. It does rely on us being able to accurately identify the animals. Here, we apply our knowledge of prior basic biology in developing the model. As an analogy, suppose we find a crocodile where we know there are no rivers or lake. We will then turn to the photograph (MCD file and H&E sections – all uploaded onto our MDV viewer) to ask if this was actually a crocodile and develop/refine/revise the annotation a little more. Eventually we will use machine learning to help us to refine these tasks but, at this point, we have painstakingly checked every predicted type of animal against the photographs of the park to ensure they have been identified correctly.

We have added the first paragraph of this response to the starting paragraph of the Discussion in order to better frame the utility of our spatial analytical approach.

5. **In figure 5, the authors investigate the occurrence of direct cell-cell neighbourhoods, instead of a 20µm radius in figure 3. This is an important question, however, although it is a bit difficult to compare due to the different visualisation, I am not convinced that there is much difference between the results of these two, as 20µm is already very close. Indeed, the same interactions are found. Can the authors highlight what this adds and why is this analysed with a completely different approach rather than using the approach of figure 3 and minimising the radius.**

Yes, the results have to be similar because for the direct cell/cell analyses (CCN) used in Figure 5, we ask the following - of the pairs of cell types which are co-located within 20µm of each other [ie $g(r=20) > 1$ in the cross-PCF analyses], which cell pairs are also in direct physical contact? To provide greater clarity, please see the diagram below (Figure R7):

Cross PCF analysis asks if there is a greater number of cells of all types (in this case cells of type B) around a 20 μm radius from the centroid of Cell A [for $g(r=20)$]. Calculation of cross-PCF is performed using the cell centroids of a pair of cells. The 'cell centroid' is a point in the centre of the nucleus of the cell.

ACN or adjacency cell network analysis asks if cell pairs that are found co-located with each other [$g(r=20)>1$] is also in physical contact with each other. To do this it takes the segmentation masks of all cells for these pairs of cell types, inflates the edge by a pre-determined 5 μm and asks if their edges touch each other.

Figure R7. Figures show how cross-PCF is related to ACN, and the additional information it provides. CCN (the final analysis used in Figure 5) is derived from ACN as follows - for each cell type pair identified as contacting in ACN (e.g., cell types A and B), the proportion of type A cells that is in contact with at least one of cell type B is calculated, and then the z-score. A p value is computed for the z-score associated with each pair of cell types. The data are then filtered to retain pairs of cell types with $g(r=20)>1$, positive z-scores and $p < 0.05$ to create a network in which each connection ('edge') between cell pairs encompasses these three calculations.

With regards to how the ACN and CCN add value to the cross-PCF:

The cross-PCF approach taken in Fig 3 considers cell centres only, ignoring cell shape. Advantages of this approach are the ability to compare the distribution of cell centres directly against a null distribution (complete spatial randomness) and its simplicity to compute. However, this approach has limitations. Two cell centres within short range (20 μm) may belong to cells which are not in physical contact. Additionally, cells with unusual (eg convoluted/dendritic) shapes may be in contact but their centroids may be more than 20 μm apart. ACN and CCN use the segmentation mask to ensure that the cells considered are in direct contact. We expect the two complementary methods to yield similar results, as the reviewer notes; by using both methods, rather than limiting our analysis to a single method, we can be more confident in our biological conclusions.

We have added parts of this response and Figure R7 to the Discussion and as Supplemental Information under "Extended description on role of different spatial analytical methods" and linked this with Figure 5.

6. "The pattern of spatial co-location changed at the leading edge of disease (Intm), where CD206hi macrophages appear to be the central connector for the ABI and immune cell communities (Fig 5E), linking the entire ABI network to the immune network". This sentence, and the visualization in 5d-f are a bit strong from what can be taken from the data. The data merely shows that interactions between cell 1 and 2, cell 2 and 3, cell 3 and 4 etc occur, not that there is one mega ABI-immune network.

Yes, we agree. The method we used shows that cell types 1 and 2; cell types 2 and 3, etc are found co-located with each other are immune cells (one spatial network) while another network comprise cell types 4 and 5; cell types 6 and 7 etc which are structural cells. Since

the 'interactions' are spatial co-locations and if cells of type 1 are spatially co-located with cells of type 2, and 2 with 3; then by inference, cell type 1 is spatially linked to cell type 3. However, you are right that this is not necessarily a network but spatial proximity and adjacency. We have softened the sentence and depicted this as spatial association rather than a network.

Minor comments

1. Affiliation 6 is not linked to an author

Thank you – we have corrected this

2. temporal-spatial niche instead of temporo

OK

3. throughout the text, mm is used instead of μm

Apologies – we will correct this. Font change issues

4. co-localised instead of co-located

We respectfully disagree: 'co-located' is the correct mathematical term to use, according to the mathematicians in the team.

Reviewer 2

Weeratunga and colleagues, used a 33-plex single cell imaging mass cytometry to characterize cell populations in lung samples from 7 patients with end stage IPF (obtained at lung transplantation) and 2 controls (donors), focusing on epithelial cells and immune cells. They assessed 3 areas per patient. Then, they reanalyzed publicly available scRNAseq data to determine whether ligands-receptors could be identified.

1-I do not understand why the authors suggest that alveolar metaplasia is a feature of alveolar regeneration. I would suggest instead to be purely descriptive in the results section and call this alveolar metaplasia and discuss later whether this is alveolar regeneration.

Thank you for this suggestion. We agree that, by definition, metaplasia means change from the expected cell type to the one observed. However, type II alveolar metaplasia, which is the case here (i.e. not squamous metaplasia), is well established in lung biology and histology as a post injury alveolar regenerative response. Our specialist lung histopathologists were keen to point this out and cite this paper as the original source of this widely held view² where the authors show that injury of the alveolar lining with Bleomycin led to formation of type II alveolar metaplastic cells. - Bleomycin-induced injury and metaplasia of alveolar type 2 cells. Relationship of cellular responses to drug presence in the lung. I Y Adamson, D H Bowden. Am J Pathol.1979; 96(2):531–544.

Bridgit Hogan's group has also shown that type II alveolar epithelium undergo clonal proliferation after targeted injury by using lineage tracing³.

We have added these references for equating alveolar metaplasia with alveolar regeneration from the outset in the M/S and soften the assertion.

2-A major issue in this study is the way the authors analyzed the samples. They identified samples as Early/End stage/intermediate on the basis of morphology. However, when looking at figure 1I, it appears that what the authors call "Early" is already very severe disease since one cannot recognize any normal alveolar structure. This is probably why the authors did not find a difference in the overall proportion, distribution, or abundance of immune and epithelial cells in the three categories of lesions. This is a major flaw to this study.

We respectfully disagree with this, although we understand why you have come to this conclusion.

All samples were taken from lungs which were already at an advanced stage of disease as they were from patients undergoing lung transplantation. We mention this in the M/S. Early, intm and advanced are relative terms.

It is well established that the IPF disease progress inwards from subpleural regions of the lung and upwards from basal regions of the lung⁴.

In our study, progression from severe to less severe fibrosis is identifiable macroscopically (by gross anatomy and digital manipulation of the lung tissue in the explanted lung) (Figure 1B in paper, copy and pasted here for convenience – Figure R8). The biopsies were taken from earlier stages of disease defined visually (greyer where there is more advanced fibrosis) and by finger manipulation (stiffer where there is more fibrosis and more pliable where the lungs are less fibrotic).

Figure R8 (from Figure 1B in M/S) demonstrating areas in the lung from which biopsies were obtained for the different disease stages

In addition, two senior pulmonologists and a senior histopathologist (with >20 years specialist experience) examined the H&E section and determined the formal histopathological characteristics of these ROIs. These were tabulated in Suppl Table 2 which provides the amount of fibroblastic foci (feature of earlier disease), bronchiolisation, smooth muscle hyperplasia and collagen (feature of later/more advanced disease). The lung sections were then categorised into early, mid and advanced by the histopathologist independent of the knowledge of where these were taken from the lungs. This histopathology analysis of the 58 ROIs were then compared to the original macroscopic classification (performed by an independent pathologist) of early, intm or advanced. All but two of the ROIs were called concordantly with the macroscopic classification by the two pathologists, in keeping with established histopathology definition of usual interstitial pattern (UIP) pattern fibrosis in IPF lungs⁴.

Hence, there is a heavily invested pre-analysis classification of the ROIs into different stages of disease. Finding the same proportion of immune cells in different stages of disease is a major result. It suggests that what changes as disease advances is not the composition of immune cells but, potentially, how they behave and their interaction with each other and with the epithelial cells.

We trust that this is clearer, and have incorporated this point in the Results section. We have also made it clearer that early, intm and advanced are used as relative terms in a diseased lung. Thank you for pointing this out.

3-The authors identified a population of epithelial cells with low expression of CD11c and CD45. This looks like an artifact were myeloid cells in close proximity to epithelial cells are considered as “positive” by the system. What is the evidence for the actual existence of these cells? Do they identify these cells also scRNAseq data bases?

These are indeed epithelial cells that are adjacent to myeloid cells and the cluster is annotated as 'ABI_b DC ADJ' cells i.e. ABI-b (epithelial cells) DC(myeloid cells) and ADJ(adjacent or in close proximity). 'Adjacent cells' is a feature in tissue single cell mass cytometry that is not seen in suspension single cell flow cytometry and CYTOF methods, and single cell RNA sequencing as these latter methods do not have the physically fixed locations as in embedded tissues. Thus, tissue analysis uniquely provides the spatial association of cells with each other. Since we do not have separation of single cells as we do in droplet single cell RNA sequencing or suspension single cell CYTOF or FACS, we have to computationally segment the cells in situ.

A computational analytical algorithm (Deep Cell)¹ first segments the cells, identifying the nucleus as the centre of a cell, and using the cytoplasmic and cell surface markers to map the rest of each cell. Segmentation is performed as shown in Figure R9A. Edges of unusually shaped cells that are captured without nuclei (see Figure R9A) are discarded computationally. The average intensity of each marker for each cell is then obtained and submitted for unbiased dimensionality reduction and clustering process with R – Phenograph. All samples are integrated and batch correction is performed (Harmony), and finally, an overall representation of the marker expression per cluster across all samples (expression density histogram) is then generated (Figure R9B-C). High number of single cells (n= 313,000) helps reduce the contribution of signals due to technical variabilities; and consistent computationally-applied rules on which cells to keep and discard across all cells and samples provides an unbiased final single cell landscape.

C.

Figure R9. A. Diagram demonstrating the process of single cell segmentation by DeepCell B-C. UMAP showing final annotated clusters and scaled marker expression density plots for selected cell clusters

Final marker expression decisions are taken in conjunction with biological understanding. The judgement of whether the expression of any one marker is absent, high or low is not made visually but by using an unbiased method of calling the expression based on the expression density histogram of each marker on the cell cluster. We performed iterative calling of negative, positive, high or low of each marker by a blinded method on at least three different occasions and verified that these were reproducible and biologically meaningful before fixing the final 'call' of expression of each marker for each cell cluster. An important iterative step after this involves checking the annotated cell cluster with the tissue via the MCD file and H&E sections to ask if what we have called via the expression density histogram is what we see in tissue. We then check every annotated cell cluster against the MCD files and H&E sections to ensure they have been identified correctly. Once this is set, we do not adjust the call in order to make this a comprehensive and unbiased method of calling of the staining.

Figure R10. Original Suppl Figure 1 showing annotation steps

As can be seen from the annotation steps in Figure R10, cell clusters that appear to be artefactual, are either filtered or not 'called'. In the case of 'epithelial cells with CD11c and CD45 expression' we show that this is a cell cluster where DC (CD45+CD11c+ cells) are found adjacent or in immediate close proximity to the epithelial cells. We showed this in Supl Figure 2A; cut and pasted here for convenience (Figure R11):

Figure R11. MCD antibody expression image (top) and corresponding cell centroid plot (below) showing the antibody staining for ABI_b DC ADJ cell type in the top image and the different cell types together in the cell centroid plot (bottom image).

Cells that are less established biologically eg the aberrant epithelial cells in alveoli, are manually cross-checked with scRNAseq data. This exercise was shown in Suppl Figure 3.

We have added a modified text of this response in the Methods and also as Supplemental Information under the heading 'Expanded information on tissue harvest, preparation and analysis of signal expression'. Thank you very much for raising this point.

4-The authors observed correlations between ABI and immune cell infiltration, with a statistical link suggesting colocalization between ABI and CD206mid macrophages, CD206hi macrophages and CD103+ CD4 T cells. More specifically, they suggest that CD206hi macrophages were in close contact with ABI_b-DC ADJ and CD206mid macrophages with ABI_a cells. Their in silico analysis suggest that fibronectin and MIF are potential ligands in this interaction. They should provide experimental data for a communication between macrophages and epithelial cells, through these ligands, driving the differentiation of epithelial cells.

With respect, we think this is beyond the scope of this paper. This paper has been a very large project where we have validated a 33-plex panel of antibodies in a tissue sample cut from a paraffin embedded block. This has taken nearly two years and represents a first of its kind where nearly all single cells in a region of tissue are identified and novel mathematical methods are compiled to answer a specific question in lung fibrosis and regeneration. We applied our clinical, biological and immunology knowledge to ensure the most accurate result in annotation and interpretation of findings, to address our aim which is to define the temporo-spatial landscape of the regenerating alveolar niche in IPF. The data represent the first substantive spatial analysis of its kind in the regenerating niche in IPF, using protein markers. Our next step is indeed to test this in murine deletion models and human organoids with CRISPR methodologies. To do this properly and substantively, will take another 2-3 years of work.

Reviewer #3 (Remarks to the Author):

Weeratunga and colleagues take a single-cell spatial proteomic approach to explore the temporo-spatial cell atlas of the regenerating alveolar niches. Their extensive efforts have produced a high-resolution map from well-characterized and registered idiopathic pulmonary fibrosis (IPF) samples from different disease stages. By conducting comprehensive analyses and thorough comparisons, the authors have identified CD206hi alveolar macrophages as central immune cells in the immune-alveolar epithelial interactome.

While I have a generally positive view of the manuscript, several issues should be addressed. The reproducibility of the single-cell data analysis across donors/conditions needs further attention, additional assessments are required to evaluate the robustness of the cell clustering and annotation, and there are missed opportunities for more thoroughly characterizing receptor-ligand cell communities in spatial space. Furthermore, the authors should refine their statements and conclusions to ensure they are presented with greater precision and rigor. In addition, the following points must be addressed to justify publication in Nature Communications:

Thank you very much for reading the M/S and your very helpful comments and critique. We have addressed these point-by-point below.

Major comments:

1. Please provide detailed quality control for each sample/experiment. For example, an average number of antibody signals per cell, any potential low-quality cells were removed, what criteria were used, and how the cutoffs were determined.

Thank you for this comment.

Please find the following to provide context before describing our QC approach:

Since there are only 31 markers (antibodies) (and 2 nucleus markers) in this single cell IMC (sciMC) [in contrast to single cell RNA sequencing (scRNAseq) data where there are up to 20,000 markers (genes)], our approach to QC is different from that used for scRNAseq. The 31 antibodies are chosen very carefully to answer the question posed (i.e. how immune cells behave temporally and spatially in regenerating alveolar niche in IPF). These antibodies are well established and are 'identifier' or unique phenotyping markers with high specificity for the cell type, and good binding properties. Selection of antibodies and optimisation of marker expression are done carefully before the experiment so that, at the point of QC for the expression data, very little data are filtered, unlike for scRNAseq. Biological knowledge is highly involved in analysis e.g. antibodies reflect prior biological knowledge in contrast to scRNAseq where transcriptome-wide markers allow greater scope for discovery.

We describe below this pre-experiment optimisation method, and then outline the QC steps, as requested.

Antibody validation, sample staining and acquisition of expression

Lanthanide metal conjugation of antibodies was validated by first checking the recovery of antibody post-conjugation. Then, we checked for successful metal conjugation by binding the antibody to iridium labelled antibody capture beads AbC™ Total Antibody Compensation Beads (Thermo Fisher, USA, CAT#A10513) and acquiring on a Helios system (cell suspension mode). Finally, we checked that the antibody had refolded and retained the ability to recognize

antigen by using the post-conjugation antibody in either a two-layer immunofluorescence (IF) with a fluorescently-labelled secondary antibody recognizing the primary antibody species or directly by IMC using the Hyperion imaging module (tissue mode).

All antibodies were validated in healthy AND disease (IPF) lung and other organ sections (prostate, kidney), and healthy secondary lymphoid tissue (tonsils) and shown to work well in the Hyperion system at pH9 using the Heat Induced Epitope Retrieval (HIER) retrieval system. The antibodies were further validated for use by single-plex immunofluorescence staining of lung sections from the same cohort. Positive tissue controls and no primary antibody controls were used in each instance (examples shown in Figure R12). Success of the antibodies was determined by a cross functional team including specialist pathologists.

Figure R12. Immunofluorescence staining of tonsil (left) and lung (right) tissue with exemplar antibodies before testing with metal tagged antibodies in IMC.

An important step in QC is the systematic correction for any signal spill-over between excitation channels. This is typically very small in mass cytometry and further minimised by careful optimisation and selection of isotopes and antibody concentrations. However, spill-over can be present and can complicate interpretation of data. For example, signal crosstalk can result in incorrect identification of cells as expressing an intermediate level of a marker. In our experiment, polystyrene capture beads were single stained with each antibody used in the experiment, analyzed simultaneously in the mass cytometer. The CATALYST R/Bioconductor package (<https://bodenmillergroup.github.io/IMCDataAnalysis/spillover-correction.html>), was used to deconvolute the different bead populations, estimate spillover signal in all channels, and compensate the data⁵.

Once these QC steps had been performed, IMC staining was performed in a minimal number of batches (in our case, three), in equally balanced groups [balanced for Early, Intm and Advanced lung sections, healthy control lung sections and on the day, a positive control sample (tonsils) and negative controls were used for each batch].

Acquisition [i.e. ablation of tissue by the tissue mass cytometer (Hyperion)] was done on one machine sequentially over 2 weeks and all checks for technical stability of the machine were performed between and again, just before each acquisition. Prior to each set of ablation, the Hyperion Tissue Imager was calibrated and rigorously quality controlled to achieve reproducible sensitivity based on the detection of one of the isotope (¹⁷⁵Lutetium). In brief, a stable plasma was allowed to develop prior to ablation of a multi-element-coated “tuning slide” (available from Standard Biotoools). During this ablation, performance was standardized to an acceptable range by optimizing system parameters using the manufacturer’s “auto-tune”

application or by manual optimization of XY settings whilst monitoring ¹⁷⁵Lutetium dual counts. Small regions of tonsil tissue were first targeted to ensure complete ablation of tissue during the laser shot with ablation energies adjusted to achieve this where required.

In the following paragraphs, we provide a step-by-step method of data processing and filtering:

QC steps for expression data

- a. Expression signals from all 31 antibodies and 2 DNA markers (in the form of a TIFF file) are extracted from the imaging files (MCD) for each excitation channel.
- b. The data are first examined for spillover of signal between excitation channels – these should be less than 5%. We found no spillover due to prior optimization of antibodies and metal tag allocation to appropriate channels as described above.
- c. Expression of each antibody is visually examined in all ROIs using MCD viewer of the images, compared to H&E sections; and a ‘first pass’ list of antibody filter is performed as follows:
 - i. Antibodies which show diffuse staining in all cells without discrimination for appropriate cells or which showed very low levels of expression in all samples including positive control on day of staining, are excluded from further use. The following antibodies were excluded from analysis - BDCA2, KRT14 (see Figure R13)
 - ii. Antibodies showing no staining at all. None were excluded
- d. First pass segmentation of cells on images is performed using a computational analytical algorithm (Deep Cell). The average intensity of each marker for each cell is obtained and submitted for unbiased dimensionality reduction and clustering process with R – Phenograph. All samples are integrated, harmonized and batch corrected (using Harmony) to minimize individual sample and batch effects (as in scRNAseq analysis), and an overall representation of the marker expression per cluster across all samples (expression density histogram) is then generated (Suppl Fig 1i). Annotation as described in the paper (Suppl Fig 1a-g).
- e. At this point, three specific QC steps are undertaken:
 - i. A cell cluster that does not show any antibody expression is identified. This is usually because none of the 31 antibodies are found on the cells. These are excluded from further analysis. 7.9% of total cells from all samples fell into this category.
 - ii. Cell clusters that comprise less than 0.1% of total cells were excluded. There were 5 such cell clusters.
 - iii. A cluster is termed undefined (UD) if there is insufficient information to annotate its constituent cells and if they expressed markers which are biologically implausible e.g. defining markers in epithelial cells AND immune cells. 6 such clusters were identified, and comprised 10.8% of the total cells from all samples. These information can be found in Suppl Table 4.

Figure R13. MCD file expression images showing the expression of low quality staining (BDCA2 and KRT14) and good quality staining (KRT5, CD20 and CD206) in IMC, in two representative IPF regions of interest (upper and lower panels)

We have placed a modified version of this in the revised M/S, in Supplemental Information, under “Extended quality control (QC) steps’.

2. For each antibody marker expression, whether the distribution is consistent between samples and/or conditions. Any potential low-quality antibody markers?

Thank you for this query. We understand the question but we cannot test the distribution between samples and between conditions because a range of values are expected, due to different types of cells that express any one antibody in that sample and in that condition. For example, Sample A in the ‘Early’ condition could contain, say, 20% macrophages expressing CD206 at a low level per macrophage but in Sample B from the ‘Advanced’ condition, CD206 expression on macrophages could be very high per macrophage and, therefore, the difference in CD206 expression per cell in that sample could be due to the condition or biology rather than a technical problem.

Instead, to test consistency in antibody staining, as per our response to comment 1, we pay particular attention to pre-experiment optimisation of antibody staining quality, and then perform a final QC of expression data after staining. Antibodies and clones with optimal avidity to epitope, high specificity and isotope metal-tagging checks on actual tissues are performed to ensure consistent expression in healthy lung and lymphoid tissues.

‘Low quality’ antibodies are excluded at the pre-staining optimisation stages (see response to comment 1).

Low quality staining after the pre-staining optimisation is defined as

(a) antibody expression in cell clusters in healthy control samples which varied greater than upper and lower limits of the IQR (Figure R14)

(b) where there is poor specificity in healthy and disease samples ie antibody is expressed in cells where it should not be expressed – eg KRT14 expression on CD4 T cells in all samples (Figure R13)

(c) where there is low expression in all samples below background expression.

These checks are done using density expression histogram (Suppl Fig 1i) and visually on MCD viewer matched to corresponding H&E sections

For (a), we select cell types (i.e. cell clusters defined by annotation using all markers) where we know the protein expression typically remain stable between samples and ROIs in healthy control – e.g. CD206 in resident alveolar macrophages in healthy lungs, EPCAM in alveolar epithelial cell in healthy lungs and CD4 in T lymphocytes in healthy lungs (examples shown in Figure R14). For (b)(antibody specificity), we examined every antibody expression in every ROI in HnE section and MCD file, and excluded KRT14 due to its expression on immune cells (Figure R13), and for (c), BDCA2 was excluded from analysis (Figure R13).

Figure R14. Boxplots showing median metal intensity of CD206, CD3, CD4 and EPCAM in clusters 1, 20, and 10 for regions of interest drawn from healthy samples

3. I appreciated the detailed description of cell clustering steps. Please quantify whether the clusters are stable and robust. Whether the single or the combination of multiple markers used for annotation are specific. Any potential over-splitting or under-splitting? Is there any potential bias between batches and donors?

Thank you for this request.

To provide context - cluster stability is a well-established metric in scRNA sequencing as the large number of markers involved in generating a cluster (20K genes) means that different numbers of clusters can be generated. In scIMC, the number of markers is substantially lower (31 in our case). These markers/antibodies are key and well-established phenotypic markers which are selected to identify clusters. Although it uses the same principles, scIMC differs from suspension mass cytometry (sMC or CYTOF) because the usual function of excluding doublets is not available and cells that are adjacent to each other have to be ‘called’ on annotation. Biological input assumes a much greater importance than scRNAseq and sMC. Like sMC, clustering is more stable as we sacrifice depth of annotation for accuracy. Our 31 antibody panel is designed to identify the following immune cells accurately - NK cells, $\gamma\delta$ T cells, Tregs, CD4 T cells, CD8 T cells, macrophages, monocytes, neutrophils, B cells, but not subsets of CD4, CD8, neutrophils or NK cells for example.

For clustering stability, we use a biologically-driven approach to clustering and iterative loops of annotation (described in Suppl Fig 1) and once this is stable biologically, we check technical stability of this clustering using the ‘industry standard’ method of quantifying cluster stability in scIMC proposed by Bodenmiller - the ‘silhouette score’ and the ‘neighborhood purity score’⁷. The silhouette score measures how similar an object is to its own cluster compared to other

clusters. The score ranges from -1 to $+1$, and a high value indicates that the object is well-matched to its own cluster and poorly matched to neighbouring clusters. If most objects have high values, then the clustering configuration is appropriate. If many points have low or negative values, then the clustering configuration may have too many or too few clusters. In scIMC, the values are typically much lower than scRNAseq due to the low numbers of markers - score of 0.04 and 0.06 are suggested in the scIMC data from Bodenmiller⁷ and we use these values to check that the K selected using biologically determined clustering falls in this range (which it does; average SCE 0.04) (Figure R15).

Figure R15. Boxplots showing silhouette scores for individual clusters at a resolution of $K = 15$

The neighbourhood purity score is the percentage of cells from the neighbourhood that has the cell type label in question. A high average neighbourhood purity indicates that the cells are well clustered. For $K=15$, our average purity score is 0.88 (Bodenmiller was 0.87 to 0.89) (Figure R16)⁷.

Figure R16. Boxplots showing neighbourhood purity scores for individual clusters at a resolution of $K = 15$

With regards to ‘whether the single or combination of multiple markers used for annotation is specific’ – antibody specificity for the protein is tested pre-staining as described above. QC of expression data also helps ensure specificity. The markers are not ‘discovery level’, rather they have been selected to provide high level but specific identification of cell types. Eg CD4 T cells are marked by CD4+, CD3+, CD11b-, CD68-, CD1c-, EPCAM-FoxP3-. All the combination markers and interpretation are outlined in Suppl Table 4. Both expression and non-expression of a combination of markers are important in identification of the cell cluster/cell type.

In terms of bias of clustering between batches and donors, please see this context first: As part of the QC, we run batch correction with Harmony (see response to Comment 1). The three batches were almost equivalent, with little variance, and no cells or clusters were excluded in the final UMAP (Figure R17). Under and over-splitting are dealt with manually and biologically as described above during iterative rounds of annotation. For established cell types eg CD4+ T cells or B cells, and subsets of macrophages and monocytes, standard immunological knowledge was applied. For less established cell types eg the aberrant basal intermediates (ABI) epithelial cells, we cross-check with three other scRNAseq datasets [Habermann AC et al Sci Adv 2020 (IPF lung tissue)⁹; Adams TS et al Sci Adv 20202 (IPF lung tissue)¹⁰ and Kathiriya JJ et al Nat Cell Biol 2021(lung organoid studies)¹¹]. This is shown in Suppl Figure 3.

Figure R17. Upper panel shows unharmonised UMAP from three batches done on three different days (but same machine, settings and antibody mix). Lower panel shows the three UMAPs after harmonisation.

We have added cluster stability testing methods and results to the M/S under Methods.

4. In Fig 10, it would be nice to show 2D embedding. Whether certain cell populations be better separated by sub-clustering?

Yes of course. We have now provided this in Suppl Figure 1A as a complement and present it here for convenience (Figure R18)

Regarding ‘...whether certain cell populations be better separated by sub-clustering?’, our panel design focused on accurate annotation of major immune cell types, sacrificing depth for breadth. Thus, it identifies CD3 T cells for example, and its subcluster CD4 and CD8 T cells but not central memory CD8 T cells, cytotoxic CD8 T cells etc as there were neither the relevant antibodies to do so (as we were limited to 31, unlike scRNAseq where the whole transcriptome was available) nor was it part of the study design. The resulting clusters in the UMAP reflected our scIMC antibody panel design and we did not pursue further sub-clustering apart from where it was biologically required or obvious. For example, aberrant basal intermediate cells which are less established were subclustered according to the different combination of antibodies and cross checked manually with Kathiriya JJ et al Nat Cell Biol 2021(lung organoid studies)¹¹ and also Habermann AC et al Sci Adv 2020 (IPF lung tissue)⁹, Adams TS et al Sci Adv 20202 (IPF lung tissue)¹⁰. This was shown in Suppl Figure 3.

Figure R18. 2D UMAP showing final annotated cell clusters derived from 31 plex IMC staining

5. For cell cluster annotation, please also compare and confirm with previously published reference maps (e.g., scRNA-seq).

As there are only 31 markers which are selected as identifier markers for specified cell types (as mentioned above), the method for identification is unlike that of scRNAseq where label transfer between the new dataset and a reference map is typical. We refer and cross-check with single cell dataset as mentioned above for the novel aberrant basal intermediate cells (Suppl Figure 3) but identification of CD4 and CD8 T cell for example, are standard immunological knowledge and, hence, do not need confirmation.

Please also see response to comment 9 below.

6. Whether the correlation in Fig 2k relatively consistent across donors/ROIs?

In Figure 2K, the significant correlations shown were derived by comparing all cell types in the ROIs from all donors for a particular cell state. So, for example, in all the ROIs from all donors in Early state, the abundance of AB1_a cells correlated significantly with CD14^{lo} monocytes (r=0.66) (only significant correlations are shown in Figure 2K, pasted here for convenience as Figure R19). This is further illustrated in Figure 2L as a scatter plot, showing all ROIs within that state.

Figure R19. Cut and pasted original Figure 2K with annotation

7. In the cross-PCF analysis, could authors clarify why epithelial cell types were assigned as 'anchor cells'? Given the different abundance of cell types, I am just curious if picking other cell types as 'anchor cells' will result in the same conclusion.

Thank you for your query. The 'anchor cells' refer to our cells of interest (we have made this clearer in the M/S). ABIs are designated the 'anchor cells' as we would like to answer the question – 'what immune cells are found co-located with the alveolar regenerating cells (ABIs)'.

Mathematically, the cross-PCF is symmetric between the cells in the cell-cell pair (i.e., it does not matter whether the "anchor cell" of the A-B pair is of cell type A or B). Choosing a different "anchor cell" would of course produce results for the specified anchor cell e.g. if we assigned CD4 T cells as the anchor cell type, and asked what immune cells are found around CD4 T cells, then a different set of output would be generated.

8. If I understand correctly, the ACN analysis was construed by the cross-PCF analysis. It was nice to see the consistent results in Fig. 4A-B (epithelial cells as the anchors), but could the authors explain what additional info we can learn from the ACN results? Is the selected radius parameter ($r=20$) correspond to the distance between any two cells?

Thank you for this query. Yes, you are right! The ACN analysis is related to the cross-PCF analysis as follows – first, we identify co-located cell types [those pairs of cell types that have $g(r=20) > 1$]. To determine whether the edges of the cells touch each other, we then inflate the edge of the segmented cells by $5\ \mu\text{m}$ and ask if the pairs of cell types are then in direct physical contact. Not all cell types that are close and spatially associated are in physical contact with each other. As a corollary, cells with irregular shapes (e.g those with pseudopods or dendrites) may be in contact but their centroids may be more than 20microns apart. ACN identifies those cell pairs that are physically in contact with each other.

Mathematically, we would expect the two results to be similar, as you have noted.

We include below a schematic diagram which illustrates the cross-PCF and ACN analyses (Figure R20).

Cross PCF analysis asks if there is a greater number of cells of all types (in this case cells of type B) around a $20\ \mu\text{m}$ radius from the centroid of Cell A [for $g(r=20)$]. Calculation of cross-PCF is performed using the cell centroids of a pair of cells. The 'cell centroid' is a point in the centre of the nucleus of the cell.

ACN or adjacency cell network analysis asks if cell pairs that are found co-located with each other [$g(r=20) > 1$] is also in physical contact with each other. To do this it takes the segmentation masks of all cells for these pairs of cell types, inflates the edge by a pre-determined $5\ \mu\text{m}$ and asks if their edges touch each other.

Figure R20. Figures show how cross-PCF is related to ACN, and the additional information it provides. CCN (the final analysis used in Figure 5) is derived from ACN as follows - for each cell type pair identified as contacting in ACN (e.g., cell types A and B), the proportion of type A cells that is in contact with at least one of cell type B is calculated, and then the z-score. A p value is computed for the z-score associated with each pair of cell types. The

data are then filtered to retain pairs of cell types with $g(r=20) > 1$, positive z-scores and $p < 0.05$ to create a network in which each connection ('edge') between cell pairs encompasses these three calculations.

We have placed this in Supplemental Information under the heading 'Extended description on role of different spatial analytical methods'.

9. Can authors explore the correlation of 33 makers between RNA expression level and antibody level? Is it possible to project scRNA-seq data to the spatial space? This is a missed opportunity to check if the receptor-ligand cell communities predicted by CellChat have a higher possibility of physical contact between various cells in the spatial space.

Thank you for this comment.

A pre-requisite to projection of single cell data on to spatial space is to match cell types detected in scIMC to scRNAseq. We first attempted this by automated label transfers but these result in clearly discordant matching as they are designed for scRNAseq to scRNAseq data both with 20K markers (see Figure R21). A critical issue therefore, is the inability to 'label transfer' or 'label match' cell clusters generated by 31 markers (our protein-led scIMC) with cell clusters generated by >20K markers (scRNAseq). In addition, biologically, mRNA does not always match their protein expression, and the reviewer will know that there are some genes that are notoriously difficult to identify which are better observed on protein e.g. CD4.

We have actually attempted this before with suspension mass cytometry (CyTOF) panels which are narrower in scope and focused on a particular cell subset (Corridoni et al, 2020, Nature Med) where the CyTOF panel was designed for T cells specifically. There, the integration was still poor overall and many populations could not be identified. We concluded that fundamentally, this integration will depend on the quality and quantity of the CyTOF-RNA features that are common to both modalities. If there are low number of markers to separate specific subpopulations in the CyTOF dataset or if the efficiency of the markers is suboptimal or if there is no RNA-CyTOF corresponding pair, then these subpopulations will not be identifiable in the integration either. Our attempted automated integration between scIMC and scRNAseq using Seurat's automated label transfer supports this in scIMC too (Figure R21).

Figure R21. Resultant matching performed by automated label transfer using the Seurat package, showing significant mismatch between scIMC predicted cell cluster (x axis) and scRNAseq-predicted cell clusters (found within the bars) e.g. what is called CD4 T cells from scIMC is called a combination of T cells, smooth muscle cells, proliferating T cells and basal cells (see arrow).

Given these difficulties with automated methods of matching between scIMC and scRNAseq data, we moved to a manual, biology-led method of matching scIMC and scRNAseq cell clusters, which provides a high level link between the scIMC and scRNAseq data. We think this approach is sufficient to answer and discuss the question this reviewer posed ('...check if the receptor-ligand cell communities predicted by CellChat have a higher possibility of physical contact between various cells in the spatial space')

Using the Haberman single cell RNAseq lung dataset (also used in our paper; Figure 5), we first use a biology-centric approach to align scIMC clusters with scRNAseq clusters. First, key phenotype-defining antibodies in our scIMC panel are matched to their corresponding genes (Table R1). Then we identify the cell clusters that express these 15 genes in Habermann's scRNAseq UMAP, using feature plot overlays (Figure R21).

scIMC marker (parenthesis indicates the cell it defines)	scRNAseq corresponding gene(s)
EPCAM (epithelium)	EPCAM
KRT 5(epithelium)	KRT5
KRT 7/17(epithelium)	KRT17
CD31 (endothelium)	PECAM
proSpC (ATII)	SFTPC
CD14 (monocytes, 'mono-macs')	CD14
CD68 (macrophage)	CD68

CD206 (CD206 ^{neg-mid-hi} macrophage)	MRC1
HLA-DR (highest in myeloid cells)	HLADRA
CD1c (DC)	CD1c
CD11b (highest in myeloid cells)	ITGAM
CD3 (T cells)	CD3D/CD3E
CD8 (CD8 T cells)	CD8A
CD56 (NK cells)	NCAM1
CD20 (B cells)	MS4A1

Table R1. Genes in scRNAseq corresponding to the driver/identifier protein in scIMC

We postulated that the main structural, myeloid and immune cell clusters can be manually identified using these 15 genes. This was the case, evident from the feature plots (Figure R22A) which showed that cell clusters that expressed these genes matched the annotation shown by Habermann⁹ using the entire transcriptome (Figure R22B). This suggests that our 15 driver phenotypic antibodies are able to accurately identify the main cell types if we did this manually rather than in an automated label transfer method. These antibodies now provide the link between scRNAseq and scIMC data, providing a means for projecting scRNAseq cell clusters onto tissue space. For example, CD31 (PECAM) expression alone correctly identified endothelial cells annotated using the whole transcriptome's gene expression profile (Figure R21 and R22); so our CD31⁺ endothelial cells in scIMC (cluster 28 and 35) can be matched to the endothelial cell cluster in Habermann's dataset.

Figure R22 . Feature UMAPs of single cell data showing expression of structural markers EPCAM, KRT5, KRT17, SFTPC, PECAM, myeloid markers CD14, CD68, MRC1, HLA-DR1, CD1c and ITGAM and lymphocyte markers CD3D, CD8A, NCAM1 and MS4A1

Figure R23. UMAP representation of single cell RNAseq data from Haberman et al. showing delineated meta clusters using corresponding marker genes of the proteins included in the imaging mass cytometry panel in comparison with original annotated clusters by the authors in Haberman et al.

You will see in Figure R23B that Habermann had not annotated their ‘Mac’ and ‘Mono’ population, and in Figure 23A, with our first round of annotation, we have also sub-annotated our myeloid cells to monocytes, CD206 subsets. To do this, we subclustered the scRNAseq’s myeloid and lymphocyte clusters to identify the cell types we found in our scIMC samples. Epithelial cells had already been manually matched between scIMC and scRNAseq in our original manuscript (Suppl Figure 3). For myeloid cells sub-clustering produced 17 further cell clusters which we then annotated using the 4 genes matching the proteins in scIMC - CD14, CD68, HLA-DR and CD206(MRC), so as to identify CD206^{neg}/mid/hi macrophage, CD14^{hi} and CD14^{lo} monocytes and neutrophils (Figure R24). In the scRNAseq T cell cluster, CD103⁺ CD4 T cell were identified but not as clearly as in scIMC. We allocate the highest expressing CD103 (ITGAE) and CD4 T cell cluster as CD103⁺ CD4 T cells found in our scIMC. Tregs, CD4, CD8 and NK cell subsets were easily identified using the cluster defining genes.

The final annotated scRNAseq UMAP showing the relevant scIMC cell clusters is shown in Figure R24C.

Figure R24 – A. UMAP showing re-clustering of the myeloid and lymphocyte compartments B. Ridge plot showing gene expression of CD14, CD68, MRC1 and HLA-DRA in the myeloid subclusters, C. Final UMAP of scRNAseq clusters corresponding to scIMC clusters used for CellChat analysis.

With this scIMC-directed identification of the scRNAseq cell clusters (Figure R24E), we then performed receptor ligand communication analyses in CellChat¹². We used the ‘interaction weight’ which aggregates the cell-cell communication network by counting the number of links or summarizing the communication probability. Cell chat’s ‘interaction weight’ was used to predict the cell type pairs with top cell – cell communication probabilities in the scRNAseq data. Finally, we now able to ask if the interaction weight correlated with the physical co-location identified in our scIMC using our spatial analysis.

We found an overall poor concordance between the interaction weights predicted by CellChat for scRNAseq and the spatial association derived through our spatial analysis pipeline. However, the spatial association found in our scIMC data between ABI-b_DC ADJ cell type and CD206^{hi} macrophage was placed 7th overall amongst all 358 CellChat derived communicating pairs, and 3rd when considering only structure – immune cell pairing (Figure R25). These findings indicate an important point raised by this reviewer. With regards to ‘...check if the receptor-ligand cell communities predicted by CellChat have a higher possibility of physical contact between various cells in the spatial space’, the answer is ‘no’ the receptor-ligand cell communities predicted by CellChat do not concur or correlate with the spatial association as measured by cross-PCF [$g(r=20)$] or contacting cells (z score for CCN). Only 27% of interacting cell pairs predicted by CellChat (top 50 interaction weight) had $g(r=20) > 1$ in the Early Stage disease; 30% in Intm stage and 20% in advanced stage. For those pairs with $g(r=20) > 1$, there are no significant correlations between CellChat predicted interaction weight and $g(r=20)$ values although there is a trend for positive correlation (Figure R25C).

Figure R25. A-B. Cross PCF $g(r=20)$ on the y axis and the lower panel shows the contact Z score on the y axis. C. Correlation plots showing relationship between spatial statistics and interaction weight calculated by CellChat interactome analysis (X axis) for early, intermediate and late stage disease.

Minor comments:

1. Does the “ROI” represent the region of interest?

Yes, it does – this is stated in the M/S

2. The figure panels are labeled in lowercase in the figure but uppercase in the main text.

Apologies, and thank you – we have resolved this inconsistency.

3. Fig 2K is actually a table and text in small size, maybe change it to bar plots?

We have got a bubble plot showing the entire correlation matrix in Suppl Figure 7 but could not fit in to Figure 2, so have used the table to summarise instead. We have increased the font size of the text and linked this table to Suppl Fig 7 in the legend.

4. The text size in Fig 3c is unreadable.

Apologies, we have improved this.

5. Fig. 4C-D were not cited in the main text.

Thank you, we now cite Fig 4C-D on the right part of the text.

Reviewer #3 (Remarks on code availability):

I have checked the code, the description of the methods and instruction were detailed. The scripts for this paper's datasets may need to be provided.

LingPeiHo/Ho Taylor Byrne-SpOOx-2.0 – is the GitHub repository with all the scripts
A Zenodo repository with the raw datasets will also be made public once the paper is in press.

References

1. Greenwald, N. F. *et al.* Whole-cell segmentation of tissue images with human-level performance using large-scale data annotation and deep learning. *Nature Biotechnology* **40**, 555-565, doi:10.1038/s41587-021-01094-0 (2022).
2. Adamson, I. Y. & Bowden, D. H. Bleomycin-induced injury and metaplasia of alveolar type 2 cells. Relationship of cellular responses to drug presence in the lung. *Am J Pathol* **96**, 531-544 (1979).
3. Barkauskas, C. E. *et al.* Type 2 alveolar cells are stem cells in adult lung. *The Journal of Clinical Investigation* **123**, 3025-3036, doi:10.1172/JCI68782 (2013).
4. Raghu, G. *et al.* An official ATS/ERS/JRS/ALAT statement: idiopathic pulmonary fibrosis: evidence-based guidelines for diagnosis and management. *Am J Respir Crit Care Med* **183**, 788-824, doi:10.1164/rccm.2009-040GL (2011).
5. Chevrier, S. *et al.* Compensation of Signal Spillover in Suspension and Imaging Mass Cytometry. *Cell Systems* **6**, 612-620.e615, doi:10.1016/j.cels.2018.02.010 (2018).
6. Hunter, B. *et al.* OPTIMAL: An OPTimized Imaging Mass cytometry AnaLysis framework for benchmarking segmentation and data exploration. *Cytometry Part A* **105**, 36-53, doi:<https://doi.org/10.1002/cyto.a.24803> (2024).
7. Windhager, J. *et al.* An end-to-end workflow for multiplexed image processing and analysis. *Nature Protocols* **18**, 3565-3613, doi:10.1038/s41596-023-00881-0 (2023).

- 8 Korsunsky, I. *et al.* Fast, sensitive and accurate integration of single-cell data with Harmony. *Nat Methods* **16**, 1289-1296, doi:10.1038/s41592-019-0619-0 (2019).
- 9 Habermann, A. C. *et al.* Single-cell RNA sequencing reveals profibrotic roles of distinct epithelial and mesenchymal lineages in pulmonary fibrosis. *Sci Adv* **6**, eaba1972, doi:10.1126/sciadv.aba1972 (2020).
- 10 Adams, T. S. *et al.* Single-cell RNA-seq reveals ectopic and aberrant lung-resident cell populations in idiopathic pulmonary fibrosis. *Sci Adv* **6**, eaba1983, doi:10.1126/sciadv.aba1983 (2020).
- 11 Kathiriya, J. J. *et al.* Human alveolar type 2 epithelium transdifferentiates into metaplastic KRT5+ basal cells. *Nature Cell Biology* **24**, 10-23, doi:10.1038/s41556-021-00809-4 (2022).
- 12 Jin, S. *et al.* Inference and analysis of cell-cell communication using CellChat. *Nature Communications* **12**, 1088, doi:10.1038/s41467-021-21246-9 (2021).

REVIEWER COMMENTS

Reviewer #1 (Remarks to the Author):

I would like to thank the authors for their thorough response and my questions have been answered sufficiently.

Thank you

Reviewer #3 (Remarks to the Author):

Overall, I greatly appreciate authors detailed explanation, extensive analyses, and attempt to align with scRNA-seq and test the receptor ligand communication in an independent dataset. The authors had done substantial work to address most of my concerns by thoroughly benchmarking, revising the manuscript considerably, and including detailed descriptions. The other reviewers of this manuscript had some insightful questions that the authors have also addressed. I enjoyed reading this round of revision. However, I would recommend the authors make additional test and/or modeling to make sure those immune cell populations are robustly identified. A few concerns are detailed as follows:

1. I appreciated the detailed explanation of quality control steps.

Thank you

2. Sorry about the potential misleading. Could you test if the antibody marker expressions are consistent across samples for each condition, say the early stage. In other words, do you see any potential bias from one sample, a field of view, which should be excluded for downstream analyses?

Biologically, we expect differences in expression of some markers even for samples from the same stage. This is due to disease heterogeneity amongst any one stage of the tissue, a typical feature for IPF tissue. Thus, Sample A in the early stage can have slightly more CD206^{hi} AMs than CD206^{lo} AMs, compared to Sample B in the early stage; so the CD206 expression could be different.

The most reliable way of addressing this is really via the expression of antibody for relevant cell clusters in ROIs from healthy controls where we expect no or low biological variation, which we have reported in the previous response to reviewers, showing that there are no technical outliers.

We are happy to present the antibody expression density curves for samples within Early condition as requested, which was part of the QC work we do for all staining (Figure RR1). These are really only useful in detecting sample or ROI where there is suboptimal or loss of staining in the entire antibody panel (staining is performed using the cocktail of all the antibodies, not individual antibodies) or if there is no expression of the antibody for all the samples e.g. when a catastrophic technical issue has occurred, say, one antibody was not added to the cocktail. Detection of poor antibody quality e.g. KRT14 and BDCA2 has already been described in the previous revision.

Early

Figure RR1. Histogram for antibody expression in all ROIs from Early disease stage.

3. First of all, I don't believe the fewer number of features (31 antibodies) is an issue for robustness testing. If the selected features are truly informative and of good quality, only two features can still separate clusters well. Second, while I understand that the overall silhouette score and neighborhood purity score can achieve the recommended cutoffs on average, certain cell clusters might still face potential over-, under-splitting or could be of lower quality. For example, cluster #24 has a lower silhouette score but a higher neighborhood purity score—should I interpret this as a low-quality cluster that potentially contains a mixture of cells from multiple cell types? Third, the cell type annotation for each cell cluster in Fig. R15&16 is not very clear to me. The authors mentioned 29 clusters in the main text but showed 35 cell clusters in the rebuttal letter, which suggests they might have manually merged some clusters.

If that's the case, consider training a simple classification model (e.g., random forest, SVM) to determine if the captured signals are of high quality and can be used to distinguish the 29 cell clusters effectively. These results would provide strong supporting evidence for the cell annotation based on prior knowledge. Lastly, I totally understand that it's not easy to annotate immune cell population, but the authors should use a reasonable cell cluster resolution that is best supported by the captured signals.

All the cluster identity are shown in Suppl Table 4 (apologies this was not mentioned in the previous revision). Cluster number and biological description of the cell type in Suppl Table

4 matches the cluster number in the silhouette and neighbour purity score plots presented in the revision.

There are 29 clusters because 5 of the 35 clusters were excluded from annotation as they were less than 0.1% of total cells – this was stated in the paper. The 6th (cluster 34) only had 0.04% of total cells, and was transcriptomically very similar to Cluster 6 and was merged with Cluster 6. This was also stated in the text and the Suppl Table 4.

As requested, we have trained a simple classification model using binary random forest classification. For each classifier run, data were down-sampled to 1000 cells for each of the two groups. The data were split randomly at 70% and 30% ratio to be used as training and testing sets. High cluster accuracy is seen for all clusters indicating that our clusters are well-separated and internally consistent, such that this simple classifier can discriminate between them (Figure RR2).

Figure RR2. Plot showing accuracy for random forest classifier for each cluster.

Reviewer #4 (Remarks to the Author):

This revised manuscript presents a spatial analysis of the regenerating alveolar niche in idiopathic pulmonary fibrosis (IPF) using Imaging Mass Cytometry (IMC) and an advanced spatial analysis pipeline. The authors responded well to the comments, however, a few open questions need to be resolved.

Comment 2:

Thank you for the detailed clarification regarding the rationale for stage classification. While it is now clear that the terms “Early,” “Intermediate,” and “Advanced” are intended to represent relative stages within explanted, end-stage IPF lungs, the terminology may still be misleading to readers. To enhance clarity and avoid potential confusion, it is recommended to consider alternative descriptors such as “Less fibrotic,” “Moderately fibrotic,” and “Severely fibrotic,” or to include consistent reminders throughout the manuscript that these classifications refer to advanced-stage disease. Such adjustments would improve interpretability for a broader readership.

Thank you, agree and we emphasised this and also in the legend.

Comment 3:

Thank you for the clarification and the detailed explanation of the annotation process. While the rationale is technically sound, it may still be difficult for readers to interpret the ABI_b–DC ADJ cluster as a distinct entity without a clearer definition. In particular, Figure 2C shows that this cluster expresses CD45, and it remains unclear whether CD11c is also present. I recommend explicitly stating in the manuscript that this cluster represents a spatial designation—reflecting proximity between epithelial cells and immune cells—rather than a novel hybrid phenotype.

Thank you for pointing this out. We agree and have taken up the reviewers' recommendation.

Additionally, please provide further evidence and characterization of this cluster to help avoid potential misinterpretation.

This is done in Suppl Figure 2A

It would be helpful to clarify how spatial adjacency was operationally defined and whether segmentation overlap could have contributed to the apparent co-expression of markers typically restricted to immune cells.

Thank you - this was provided under Methods. To make this clearer, where ADJ clusters were first mentioned, we have also signposted to where this operational definition could be found.

With regards to ‘..whether segmentation overlap could have contributed to apparent co-expression of markers..’ we have added the following to the Supplemental Methods section for readers who might not be familiar with analytical methods of Deep Cell segmentation, a sophisticated platform that minimises the risk of overlap. We examined several segmentation methods and settled on Deep Cell for this reason.

The Deep Cell segmentation method leverages deep learning models [mainly convolutional neural networks (CNNs)], to accurately identify and separate individual cells in microscopy images. One of the main challenges in cell segmentation is dealing with overlapping or closely packed cells, which can lead to under-segmentation (merging multiple cells as one) or over-segmentation (splitting one cell into several).

To prevent overlapping cells from being merged and inadvertently incorporating more markers, Deep Cell segmentation uses these key strategies:

- 1. Instance segmentation models: Rather than simple semantic segmentation (classifying pixels as "cell" or "background"), Deep Cell uses instance-aware models like Mask R-CNN or U-Net with watershed post-processing. These models learn to distinguish individual cells, even when they are in close proximity or slightly overlapping.*
- 2. Boundary-aware training: The model is trained on labelled datasets that emphasize clear cell boundaries. This includes using special loss functions that penalize boundary errors, and data augmentations that expose the model to various crowding scenarios.*
- 3. Post-processing techniques: After the model predicts a cell probability map, methods like watershed transformation, distance transform, or contour detection are applied. These help split touching or slightly overlapping cells by finding their centers and propagating boundaries outward.*

4. *Probability maps or distance maps: Deep Cell often predicts not just a binary mask, but also distance-to-center maps or boundary probability maps. These help the model infer where one cell ends and another begins, even in densely packed regions.*

By combining a robust deep neural network with biologically informed post-processing, Deep Cell segmentation reliably separates overlapping cells and ensures each is uniquely identified as an instance, which is essential for downstream quantitative analysis.

Comment 4:

It is understandable that direct experimental validation of the predicted ligand–receptor interactions is beyond the scope of this study. However, the manuscript would benefit from using more cautious language when discussing these findings. Specifically, the roles of fibronectin and MIF should be described as predictions rather than confirmed mechanisms. This will help set appropriate expectations for readers and avoid overstating the biological significance of the computational results.

Yes, absolutely agree and have adjusted.